# Simultaneous spatiotemporal super-resolution and multi-parametric fluorescence microscopy

Jagadish Sankaran[1,4], Harikrushnan Balasubramanian [1,4], Wai Hoh Tang[2], Xue Wen Ng[1,3], Adrian Röllin[2] & Thorsten Wohland [1,3✉]

Super-resolution microscopy and single molecule fluorescence spectroscopy require mutually exclusive experimental strategies optimizing either temporal or spatial resolution. To achieve both, we implement a GPU-supported, camera-based measurement strategy that highly resolves spatial structures (~100 nm), temporal dynamics (~2 ms), and molecular brightness from the exact same data set. Simultaneous super-resolution of spatial and temporal details leads to an improved precision in estimating the diffusion coefficient of the actin binding polypeptide Lifeact and corrects structural artefacts. Multi-parametric analysis of epidermal growth factor receptor (EGFR) and Lifeact suggests that the domain partitioning of EGFR is primarily determined by EGFR-membrane interactions, possibly sub-resolution clustering and inter-EGFR interactions but is largely independent of EGFR-actin interactions. These results demonstrate that pixel-wise cross-correlation of parameters obtained from different techniques on the same data set enables robust physicochemical parameter estimation and provides biological knowledge that cannot be obtained from sequential measurements.

[1] Department of Biological Sciences and NUS Centre for Bio-Imaging Sciences, National University of Singapore, Singapore, Singapore. [2] Department of Statistics and Applied Probability, National University of Singapore, Singapore, Singapore. [3] Department of Chemistry, National University of Singapore, Singapore, Singapore. [4]These authors contributed equally: Jagadish Sankaran, Harikrushnan Balasubramanian. ✉email: twohland@nus.edu.sg

Full knowledge of a biological system requires not only information on its spatial structure but also its temporal dynamics. However, the acquisition of structure and dynamics requires complementary[1], often mutually exclusive optimization strategies[2]. Spatial resolution depends on the number of photons collected and thus sets a lower limit on acquisition time. Molecular dynamics requires acquisition times shorter than the dynamics of interest and thus sets an upper limit. Since these two limits, in general, do not lead to an overlap region, the combination of spatiotemporal super-resolution microscopy has remained a challenge. Attempts in the past either restricted time resolution[3,4] or concentration[5,6], required specialized instrumentation[7–10], or needed specialized sample labelling[11,12].

While simultaneous multi-parametric fluorescence detection (MFD) has been established for point-measurements[13,14], simultaneous parameter estimations in an imaging mode have been limited[15] and have been hampered by a lack of strategies that can bridge the limitations imposed by spatial and temporal resolution requirements and by the computationally expensive data evaluation procedures required to treat the large data sets. Here, we overcome these problems by acquiring images with high sensitivity and high-speed, using low laser powers at physiological concentrations with genetically encoded labels from the cell-biology fluorophore toolbox[16], using commercially available cameras and applying graphics processing unit (GPU)-based data processing. We therefore concentrate in this work on the use of standard equipment supported by computational analysis techniques that allow us to extract simultaneously high spatial and temporal resolution from single data sets, in real time.

We demonstrate this approach by using the following selected set of spectroscopy and super-resolution techniques, which, however, is not exclusive and can be extended depending on the needs of the user and the quality of that data. Imaging fluorescence correlation spectroscopy (Imaging FCS)[17–19] is a single molecule sensitive ensemble-based method that analyzes the fluorescence fluctuations at each pixel in time to yield spatially resolved diffusion maps and information on sample dynamics. We analyzed the spatial dependence of diffusion by FCS diffusion law[20] analysis for the determination of diffusion modes and the sub-resolution organization of the diffusing particles under investigation. Number and brightness (N&B)[21] analysis uses time binning and analyzes exclusively the mean and variance of the time-varying fluorescence intensity at each pixel from which concentration and brightness are estimated. A comparison of the brightness of a particle with the brightness of a monomer, with the knowledge of the probability of a fluorophore to be fluorescent, allows estimating the oligomerization state of the particle. Finally, we use computational microscopy to obtain complementary structural information (Fig. 1).

The collection of data in an imaging setup allows correlating spatial and temporal information. Here we use TIRF images to sort dynamics data, as collected by Imaging FCS, according to structural features, directly correlating structure and dynamics, and improving FCS data evaluation. The TIRF images can be further processed by deconvolution[22], or computational super-resolution techniques to increase the structural information available, as we show in the example of deconvolution, super-resolution optical fluctuation imaging (SOFI)[23] and super-resolution radial fluctuations (SRRF)[24]. SRRF[24], a computational super-resolution technique with its roots in SOFI[23] yields images resolved beyond the diffraction limit by performing a SOFI analysis on radiality stacks[25]. Finally, we can also use the dynamics data to identify artefacts in computational imaging methods, as shown here on the example of SRRF.

Using a recently published GPU-based algorithm for SRRF[24] and upgrading an existing ImageJ plugin developed in our group[19] to perform GPU-accelerated analysis of Imaging FCS, N&B and FCS diffusion law, we achieve measurements with high spatial localization and resolution (~60 nm and ~100 nm, respectively) and temporal dynamics (≤2 ms) from the exact same data within ~5 min, including measurement time, using a commercially available total internal reflection fluorescence (TIRF) microscope.

We demonstrate the utility of multi-parametric measurements to monitor the super-resolved structure and dynamics of two different biomolecules, namely Lifeact and epidermal growth factor receptor (EGFR). For this purpose, we recorded image stacks of mApple labelled EGFR (EGFR-mApple) and EGFP labelled Lifeact[26] (Lifeact-EGFP) on whole cells using EMCCD or sCMOS cameras for detection (50,000 frames at 2 ms time resolution covering areas as large as 128 × 128 pixels). For the simultaneous acquisition of the mApple and EGFP signals on two halves of one camera, we used a wavelength-based image splitter.

We investigate the localization, super-resolved structure and dynamics of Lifeact, a 17 amino-acid actin-binding peptide[26], demonstrating that spatiotemporal super-resolution can be achieved on one data set measured in one colour and that dynamics data can be used to remove artefacts in computational super-resolution images. Furthermore, we analyzed EGFR dynamics and organization and the cytoskeletal structure on CHO-K1 cell membranes showing that two-colour measurements provide additional knowledge that could not be obtained in super-resolution and dynamic measurements separately.

## Results

The acquisition of data at the experimentally best possible temporal and spatial resolution of 50,000 frames at 128 × 128 pixels at 16 bit per pixel results in files of 1.6 GB size, posing a serious computational challenge for pixel-wise analysis by super-resolution and spectroscopy approaches. We therefore employed a GPU to reduce computational times. A comparison of the time taken for calculating and fitting autocorrelation functions (ACFs), diffusion laws and N&B using a central processing unit (CPU) and GPU is shown in Supplementary Fig. 1 for varying sizes of input areas. The achievable improvement is dependent on the total number of pixels being evaluated. Below 160 pixels, GPU processing is slower due to the time required for data transfer to and from the GPU. From about 1,000 pixels onwards we get an improvement of at least a factor 10, depending somewhat on the exact operations. We achieved a maximum acceleration of a factor 38 in the case of N&B analysis of areas above 20,000 pixels (Supplementary Fig. 1).

Typical measurement times are in the order of 100 s where the sample might show photobleaching. Inclusive of the photobleaching correction[27], the processing times of N&B analysis, ACFs and FCS diffusion law on 128 × 64 × 50,000 pixel data are 99, 763 and 1374 s with CPU, and 27, 67 and 186 s with GPU evaluation, respectively. The GPU computation times are in the same order as the measurement time and can be performed even during acquisition.

Next, we optimized acquisition and evaluation parameters for the various techniques. While results do not depend on the camera, acquisition parameters need to be optimized for each camera model as they differ in pixel size, acquisition speed, and achievable signal-to-noise ratio (SNR).

**Calibration of Imaging FCS and FCS diffusion law analysis.** The measurement of molecular dynamics requires a time

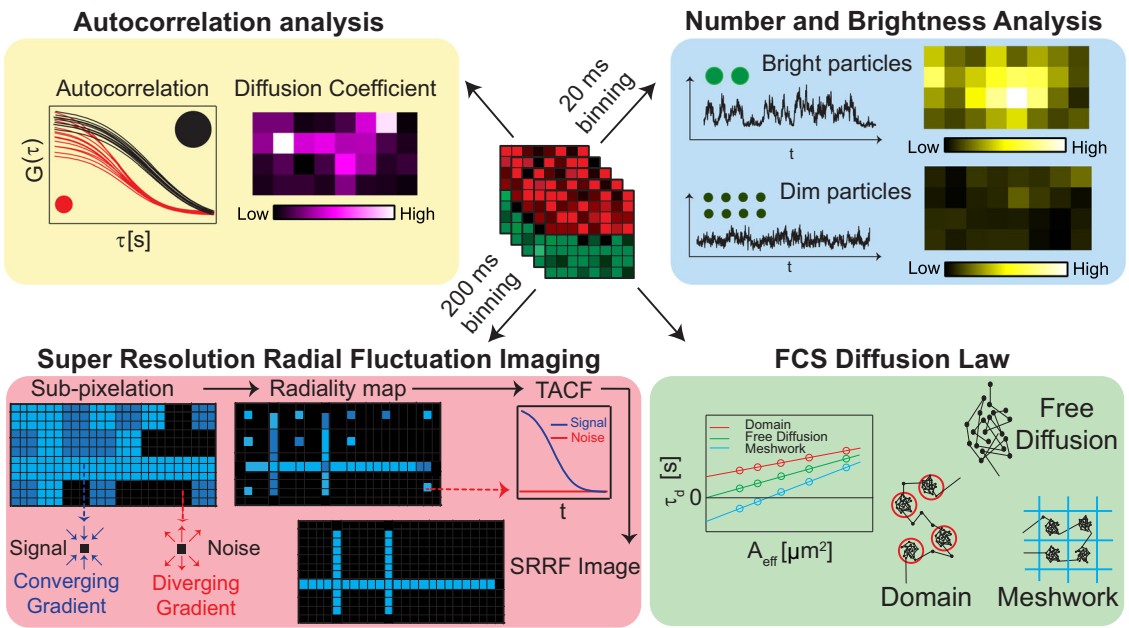

**Fig. 1 Simultaneous spatiotemporal super-resolution and multi-parametric fluorescence microscopy from a single fluorescence data set.** The various analyses performed on a single fluorescence data set are shown here: autocorrelation analysis to determine diffusion coefficient (top left), number and brightness analysis to determine particle brightness (top right), super-resolution radial fluctuations imaging to resolve structures (bottom left), and FCS diffusion law to determine sub-resolution protein organization (bottom right).

resolution about ten times faster than the characteristic process under investigation[28], in our case the time a particle needs to diffuse across the observation area of one pixel. This observation area is determined by the convolution of the pixel area and the point spread function (PSF) of the microscope[28]. The pixel size of a camera is known but the PSF must be experimentally determined[29]. This can either be done by scanning a probe small compared to the PSF over a pixel to determine the observation area[19] or it can be measured using a freely diffusing sample, e.g., a lipid probe in a one-component supported lipid bilayer (SLB), as used here[29]. For a freely diffusing particle, the diffusion coefficient ($D$) is independent of the observation area and $D$ should be constant at different spatial pixel binning. This, however, is only the case if the correct PSF is used in the calculation of the pixel observation area. Therefore, we vary the size of a Gaussian PSF until it leads to a constant $D$. In our case, the $1/e^2$ radius of the Gaussian PSF was found to be 272 and 364 nm for the measurements using 488 and 561 nm lasers, respectively (Supplementary Fig. 2). Using these lasers, the measured $D$ of lipids diffusing in a fluorescently labelled SLB consisting of 1,2-dioleoyl-sn-glycero-3-phosphocholine (DOPC) were found to be 1.91 ± 0.90 and 2.06 ± 0.39 $\mu m^2/s$ respectively (Supplementary Note 5, Supplementary Table 2), similar in range to those reported in the literature[30]. For a single pixel the observation area can now be calculated to be 0.48 $\mu m^2$ at 565 nm. With a $D$ of ~2 $\mu m^2/s$, this corresponds to an average transition time[28] of a particle through the pixel observation area of 60 ms, more than ten times slower than our recording speed of 2 ms.

The FCS diffusion law states that the average transit time through an observation area increases linearly with an increase in observation area in the case of a freely diffusing molecule, implying a zero y-intercept in a plot of transition time versus observation area[20]. Non-linearity in the diffusion law plot is reported by quantifying the y-intercept of an approximated linear function and is characterized by a non-zero y-intercept. For instance, confined diffusion leads to a positive intercept, while corralling by the cytoskeleton leads to a negative intercept. Note that this is similar to the fact used for the PSF

determination. However, the PSF calibration in principle only requires a constant $D$ within the observed spatial range while the diffusion law analysis, extrapolated to a vanishing observation area, requires free diffusion for a zero y-intercept. Here we analyzed the diffusion law in a Rhodamine PE labelled DOPC SLB from $1 \times 1$ to $5 \times 5$ binning (corresponding to 0.48 to 2.10 $\mu m^2$ with the calibrated PSF at 561 nm). The diffusion law intercepts were close to zero as expected for a freely diffusing bilayer[28] (Supplementary Fig. 2).

**Optimization of N&B parameters.** To optimize the brightness parameter, we varied time binning as well as total measurement time to determine the effects of the instrumental parameters on the estimated brightness values. Not all molecules of a fluorescent protein (FP) species are fluorescent due to incomplete maturation, misfolding, photo-bleaching and possible dark states of the fluorophore[31–33]. Hence in order to estimate the oligomerization state of a protein, one needs to estimate the proportion of FPs that are fluorescent. The proportion is estimated by computing the brightness of two different constructs, a monomeric FP and dimeric FP (a single protein consisting of two equal FPs connected by a linker and referred to as a tandem FP). We coupled the first 15 amino-acids of the RP2 protein to mApple sequences to target it to the plasma membrane[34], referred to as PMT-mApple (plasma membrane-targeted mApple) or PMT-mApple$_2$ (plasma membrane-targeted mApple-mApple).

While the dimer/monomer brightness ratio (PMT-mApple$_2$/PMT-mApple) stabilizes at 40 s total measurement time for all exposure times, it provides consistent values only above 10 ms exposure time (Supplementary Fig. 3). This is a result of the intensity thresholding (Supplementary Note 6) we use to automatically distinguish between pixels that represent the cell membrane and the background. This distinction improves with exposure time as the difference between cell and background increases with exposure. Here, we used 20 ms exposure time for N&B analysis and 100 s total measurement time for further

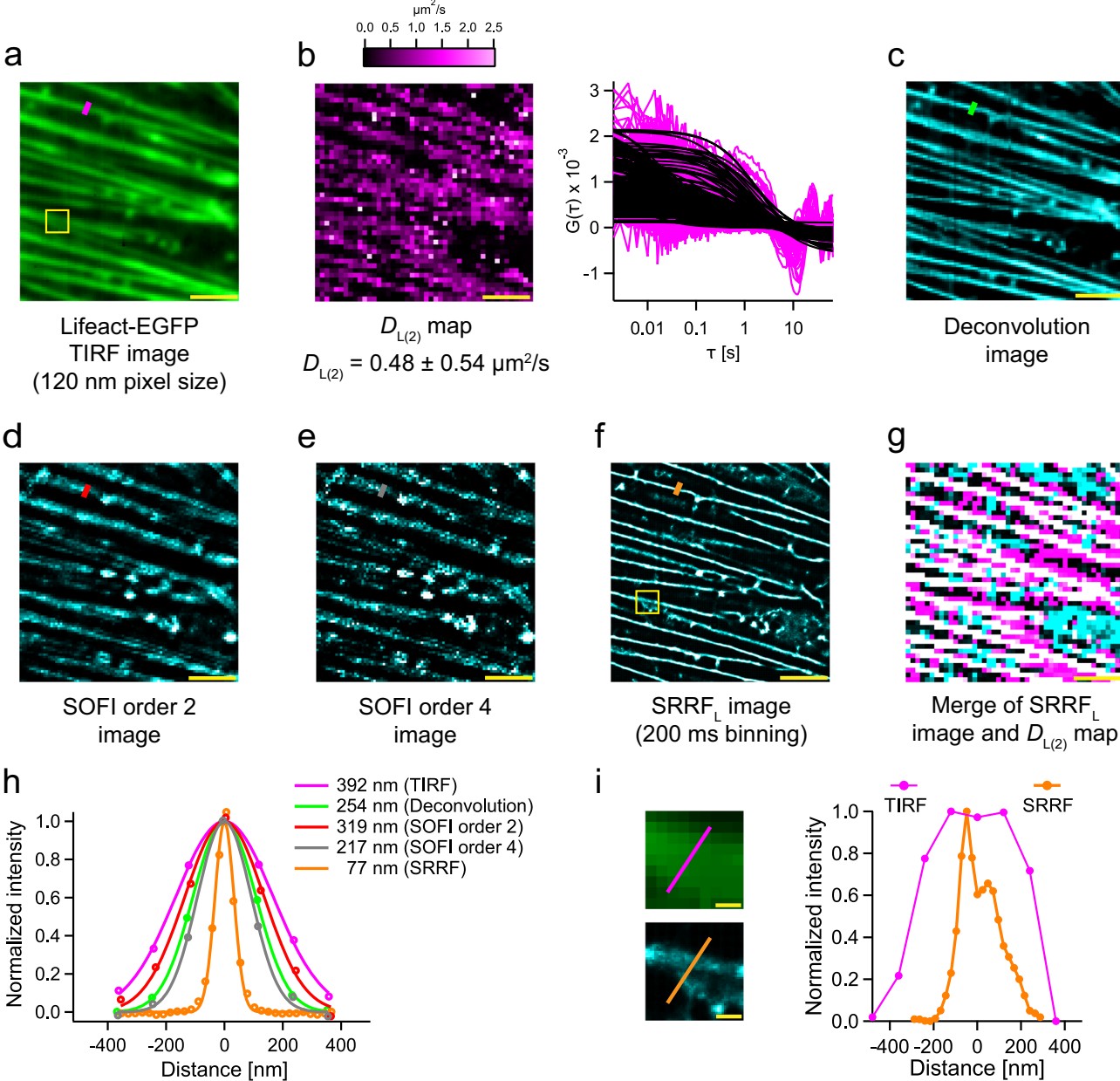

**Fig. 2 Multi-parametric analysis from a single channel fluorescence data set. a** TIRF image of CHO-K1 cell expressing Lifeact-EGFP at ×200 magnification (120 nm pixel size). **b** The $D$ map without thresholding ($D_{L(2)} = 0.48 \pm 0.54$ µm$^2$/s) is shown on the left. The COV is 113%. Representative ACFs are shown on the right. **c**–**e** Deconvolution image, SOFI image of order 2, SOFI image of order 4, respectively. **f** SRRF$_L$ image (200 ms binning; refer Supplementary Fig. 4) of the cell in (**a**). **g** Merge of the SRRF$_L$ image and $D_{L(2)}$ map. The SRRF$_L$, $D_{L(2)}$ and correlated pixels are coloured cyan, magenta and white, respectively. White pixels identify where the SRRF$_L$ image and the diffraction-limited $D_{L(2)}$ map coincide. Cyan pixels show fibres in SRRF$_L$ but no correlated $D$ is found (SRRF artefacts). Magenta pixels show a $D$ consistent with diffusion on fibres but no structure in SRRF. Due to the effect of the diffraction limit of Imaging FCS, not all pixels consistent with diffusion on fibres are observed in the SRRF$_L$ image. **h** Thickness (FWHM of the Gaussian fit) of the actin fibre in TIRF image (magenta line in (**a**)) = 392 nm. Thickness of the actin fibre after deconvolution (green line in (c)) = 254 nm. The thickness of the actin fibre after SOFI-order 2 (red line in (**d**)) = 319 nm. The thickness of the actin fibre after SOFI-order 4 (grey line in (**e**)) = 217 nm. The thickness of the actin fibre after SRRF (orange line in (**f**)) = 77 nm. Refer to Supplementary Fig. 6 for more details. **i** Enlarged views of yellow boxes in images (**a**) and (**f**). The intensity profile shows the actin fibre branching point (indicated by the orange line on the enlarged SRRF image). The peak-to-peak resolution is 96 nm at this point. The pixel sizes reported are after magnification (refer Table 3). The scale bars shown in yellow measure 2.5 µm in images (**a**)–(**g**), and 250 nm in (**i**). All values are reported as Mean ± SD. The analyses were performed on four cells from four different preparations of a single batch of cells with similar results. One representative cell is shown here. Source data is available as a Source data file.

analysis to maximize the accuracy and precision of our results. Using these experimental conditions, we found that the ratio of the brightness of the dimer to that of the monomer ($r$) is 1.55, indicating that 55 ± 1% of mApple are fluorescent (Supplementary Note 3).

**Simultaneous SRRF and FCS**. Next, we studied both structure and dynamics of actin on cells from a single data set recorded in a single wavelength channel (Fig. 2). A comparison of the actin structure, provided by TIRF and images obtained from computational microscopy (Fig. 2), and its dynamics, obtained from

Imaging FCS, enables the investigation of structure-dynamics correlations. Evaluations in this section were made on an EMCCD using ×200 magnification and $2 \times 2$ binning unless stated otherwise.

The average $D$ of Lifeact over the whole frame ($D_{L(2)}$) is $0.48 \pm 0.54$ μm²/s, corresponding to a coefficient of variation (COV—the ratio of the standard deviation to the mean) of 113% at $2 \times 2$ binning (Fig. 2b). The thickness of actin fibres as quantified by the FWHM of a Gaussian function fitted to the intensity profile in TIRF is $399 \pm 37$ nm (Fig. 2a, h). We applied three different computational super-resolution algorithms on this data set—deconvolution, SOFI and SRRF (Supplementary Figs. 4–6). For a representative fibre shown in Fig. 2, the use of second-order SOFI and deconvolution led to a 1.2 and 1.5 times improvement in FWHM (Fig. 2c, d, h), respectively. Increasing the SOFI order to four led to a further decrease in FWHM (Fig. 2e, h). SRRF processing led to a 5.1-fold improvement in FWHM. On average, the FWHM of SRRF was the lowest ($72 \pm 7$ nm; Supplementary Fig. 6). Hence, we utilized SRRF for the subsequent analysis and refer to the image as SRRF$_L$ image. The optimization of SRRF parameters is explained in Supplementary Note 7 (Supplementary Figs. 4–5).

The actin cytoskeleton exists as a meshwork in a living cell. Such a meshwork is characterized by branching at various points. Sub-resolution branching is not visible using conventional TIRF microscopy. The use of SRRF enabled the identification of branch points as quantified by the peak-to-peak distance, which was found to be $136 \pm 50$ nm (Fig. 2i). The Fourier ring correlation (FRC), which is another measure of resolution, was measured to be $90 \pm 24$ nm (Supplementary Table 3).

These measurements raise two problems. First, the diffusion coefficient has a high COV. Second, SRRF has a tendency to create artefacts similar to other computational methods[35] (see Supplementary Fig. 6 for a comparison of SRRF with SOFI and deconvolution). These artefacts show structures not detected in the original TIRF image, often related to intensity variations in the background, presumably from bright but mobile particles. We therefore used the TIRF image to identify structures and sort FCS data into fibre and non-fibre groups. The FCS data were analysed and unusually low $D$ values, not attributable to actin diffusion on fibres, are excluded further refining the map of areas where fibres exist. Finally, we utilized the resulting $D$ map to correct for artefacts in SRRF. In the following paragraphs, we explain this approach in detail (Supplementary Fig. 7). For clarity of notation we provide all parameters $P$ (diffusion coefficients, $D$, number of particles, $N$, or number of pixels, $n$) for a probe (L for Lifeact) with correction (TIRF filtered, $D$ filtered, and/or SRRF filtered), location (on or off fibres) and pixel binning (2 or 4) denoted as $P_{\text{probe, location(binning)}}^{\text{correction methods}}$.

The $D$ and SNR of the ACFs vary widely across the image. In general, we find two different cases: ACFs off fibres and ACFs on fibres. These ACFs can be differentiated by using the TIRF image as a mask by selecting a simple intensity threshold to maximize fibre retention (Fig. 3, Supplementary Fig. 7, Supplementary Note 8, Supplementary Table 4, $n_{\text{cells}} = 4$). The ACFs on fibres as judged by TIRF yield $D_{L,\text{on}(2)}^{\text{TIRF}} = 0.58 \pm 0.48$ μm²/s (83% COV) at $2 \times 2$ binning. The histogram of $D$ indicated the presence of two pools of Lifeact molecules (Fig. 3). In order to remove the effects of slowly diffusing Lifeact aggregates, we used the minimum value between the peaks as a threshold. After thresholding at 0.2 μm²/s, the diffusion coefficient is $D_{L,\text{on}(2)}^{\text{TIRF},D} = 0.77 \pm 0.43$ μm²/s (56% COV). This $D$ map, which has been filtered by TIRF, and FCS, we call the $D_{L,\text{on}(2)}^{\text{TIRF},D}$ map. Masking SRRF images by the $D_{L,\text{on}(2)}^{\text{TIRF},D}$ map removes artefacts from SRRF, by keeping only those SRRF

pixels that possess a $D$ consistent with diffusion on fibres (Supplementary Fig. 8).

The areas off fibres could not be fit at $2 \times 2$ binning due to low SNR (Supplementary Figs. 9-box 4, 10 and 11). At $4 \times 4$ binning most of the ACFs off fibres exhibited two diffusion components, a fast ($D_{L,\text{off}(4)}^{\text{TIRF}}$; Supplementary Table 5), and a slow ill-defined component. The slow ill-defined component varies widely and randomly (0.01–0.1 μm²/s) and is probably an artefact of slow measurement system fluctuations or sample movement, unrelated to actin diffusion, as can sometimes be seen in FCS at very low amplitudes[36]. We do not further evaluate this slow component.

The fast component ($D_{L,\text{off}(4)}^{\text{TIRF}} = 4.02 \pm 3.98$ μm²/s; off-fibre, Supplementary Fig. 9-box 8) was isolated by calculating and fitting the ACF only at short lagtimes τ between 2 ms and 0.5 s (Supplementary Figs. 10 and 11). This part of the ACF has a high standard deviation, as the time resolution of the camera is only 2 ms and thus does not capture the full ACF for the fast-moving particles. However, the fast component between the pixels indicates rapidly diffusing particles (2.7 fold faster than those on fibres-$D_{L,\text{on}(4)}^{\text{TIRF}} = 1.47 \pm 0.99$ μm²/s) which could be free Lifeact-EGFP or Lifeact-EGFP bound to G-actin. Running the camera at faster frame rates would allow fully capturing faster diffusing particles but at the same time limits the field of view as currently fast camera read-out is only possible with smaller regions of interest. The faster $D_{L,\text{on}(4)}^{\text{TIRF},D}$ compared to $D_{L,\text{on}(2)}^{\text{TIRF},D}$ binning results from larger contributions from particles diffusing off fibres when using the larger $4 \times 4$ area for correlation analysis.

Finally, the SNR in Imaging FCS is highest (i.e. the ACF amplitude is highest and number of particles $N$ is lowest) on pixels on fibres[37] as determined by SRRF (Supplementary Table 5 and Supplementary Fig. 12). Pixels that are identified to be off fibres by SRRF but would qualify as on fibres by TIRF show a correspondingly lower SNR, demonstrating the consistency between structural and dynamics data.

**Two-colour combined FCS and SRRF.** The full extent of the information however is gained when analyzing both wavelength channels. To investigate the interrelation between the actin cytoskeleton structure, which is relatively static on the time scale of our measurements, and EGFR mobility and organization, which is highly dynamic, we optimized the signals by spatio-temporal binning for either FCS and FCS diffusion law, N&B (red channel, Fig. 4a–d, Supplementary Fig. 13) or SRRF (green channel, Fig. 4e–g) analysis. SRRF analysis led to an improvement of the FWHM of the Gaussian fit to the intensity profile across actin fibres from 465 to 130 nm. The FWHM is not as low as that from single channel Lifeact experiments (77 nm) since a larger pixel size of 240 nm was used in the dual-channel measurements.

The diffusion coefficient of EGFR ($D_E$) was found to be $0.19 \pm 0.15$ μm²/s with an FCS diffusion law intercept of 2.58 s, indicating intermittent trapping of EGFR (Table 1, Fig. 4). N&B analysis of EGFR-mApple showed an intermediate brightness ($B_E$) between PMT-mApple and PMT-mApple₂, indicating that it contains a mixture of EGFR monomers and at least dimers (Table 1, Supplementary Fig. 13). The average brightness corresponds to an $r$ value of 1.35 which translates to $63 \pm 3\%$ dimers. The large error associated with the $D_E$ is attributed to the inherent heterogeneity in the cell membrane. The various factors contributing to the heterogeneity are investigated using a multi-parametric analysis.

EGFR-mApple expressing cells show both regions of uniform intensity and regions exhibiting visible clustering (Fig. 4a). Analysis of the regions of homogeneous brightness in the mApple channel by Imaging FCS and the FCS diffusion law

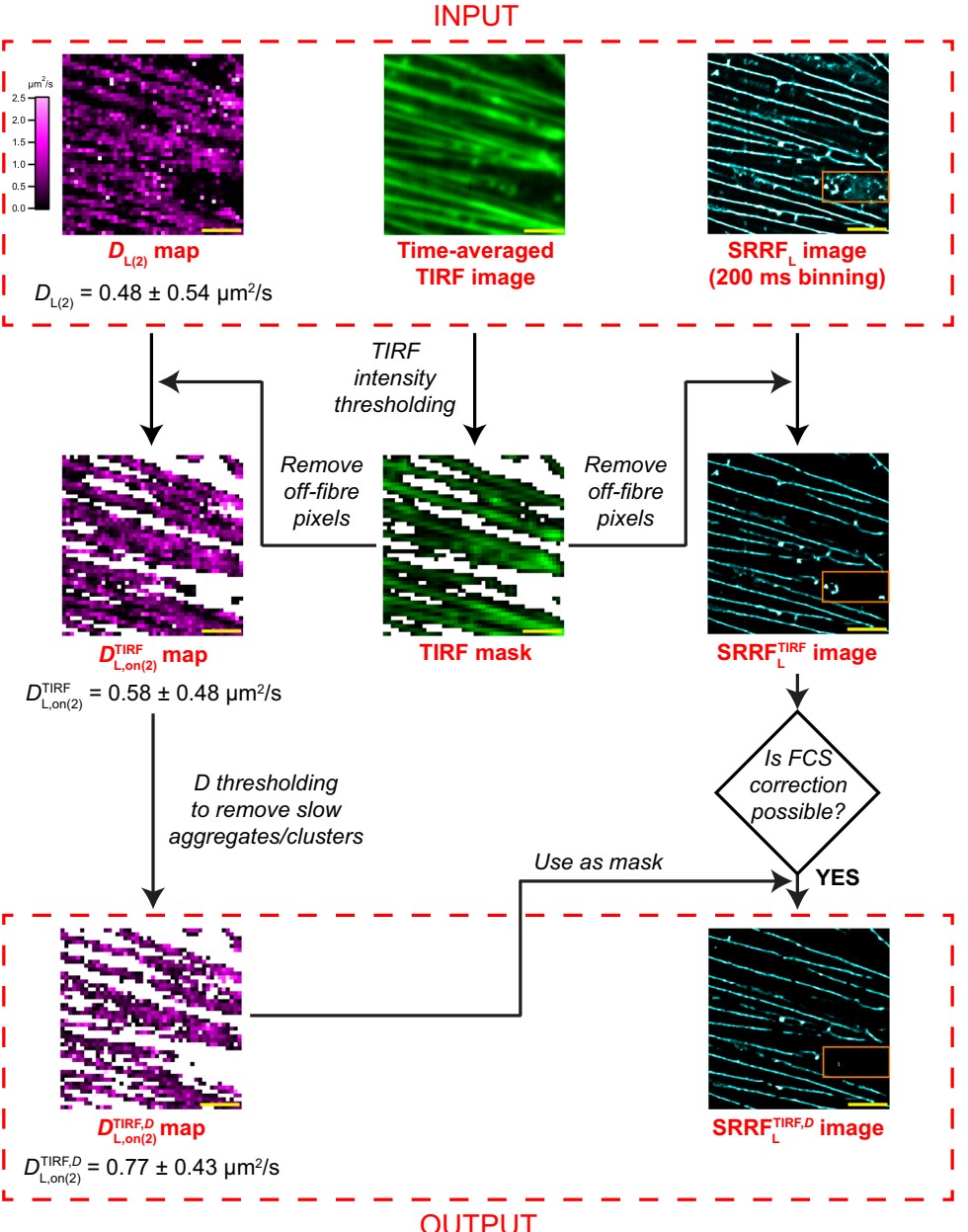

**Fig. 3 Schematic outlining the strategy to improve the accuracy and precision of the *D* map, and its further use to correct the SRRF image.** The untreated $D_{L(2)}$ map, TIRF image and SRRF image of a CHO-K1 cell expressing Lifeact-EGFP serve as the starting files. First, we create an intensity filtered TIRF mask which is then applied to the $D_{L(2)}$ map and SRRF image to remove the contributions from off-fibre artefacts generating the $D_{L,on(2)}^{TIRF}$ and $SRRF_{L}^{TIRF}$ images respectively. The $D_{L,on(2)}^{TIRF}$ map is then filtered by thresholding $D$ values to remove the slow clusters. The resulting $D_{L,on(2)}^{TIRF}$ map is used as a mask to remove artefacts from the $SRRF_{L}^{TIRF}$ image to generate the $SRRF_{L}^{TIRF,D}$ image. For the dual-colour measurements, the Lifeact SRRF image is filtered only by using the TIRF image as a mask. The orange box indicates an area that contained artefacts in the original $SRRF_{L}$ image and are now removed in the $SRRF_{L}^{TIRF,D}$ image.

resulted in $D_E = 0.24 \pm 0.16 \, \mu m^2/s$ and a positive diffusion law intercept of 1.81 s (Supplementary Fig. 14), indicating transiently trapped diffusion and possibly EGFR oligomerization or clustering beyond the resolution of our system[38]. The existence of oligomers is corroborated by N&B analysis of the same regions. EGFR-mApple showed a brightness intermediate between monomers and dimers (Supplementary Fig. 13), suggesting that $18 \pm 6\%$ of receptors are found in dimers, although we cannot exclude the existence of small amounts of higher oligomers[39].

When analyzing the regions with visible clusters present, we obtain a $D_E = 0.11 \pm 0.08 \, \mu m^2/s$ (Supplementary Fig. 14) via FCS.

In the case of N&B, the ratio between EGFR brightness to that of the monomer ($r$) was found to be 1.6. In this case, a simple dimer model is not reasonable anymore.

Given that only 55% of mApple molecules are fluorescent based on calibration experiments, $r > 1.55$ indicates the presence of oligomers larger than dimers (Supplementary Note 12). For instance, the value of $r = 1.6$ would translate into 71% monomers and 29% trimers, or 94% dimers and 6% trimers. It is important to note that if more than two species are present, extra experimentally measured brightness values are necessary to resolve the extra species. Nevertheless, the equations provide information on the existence of higher oligomers and determine limits on their fractions.

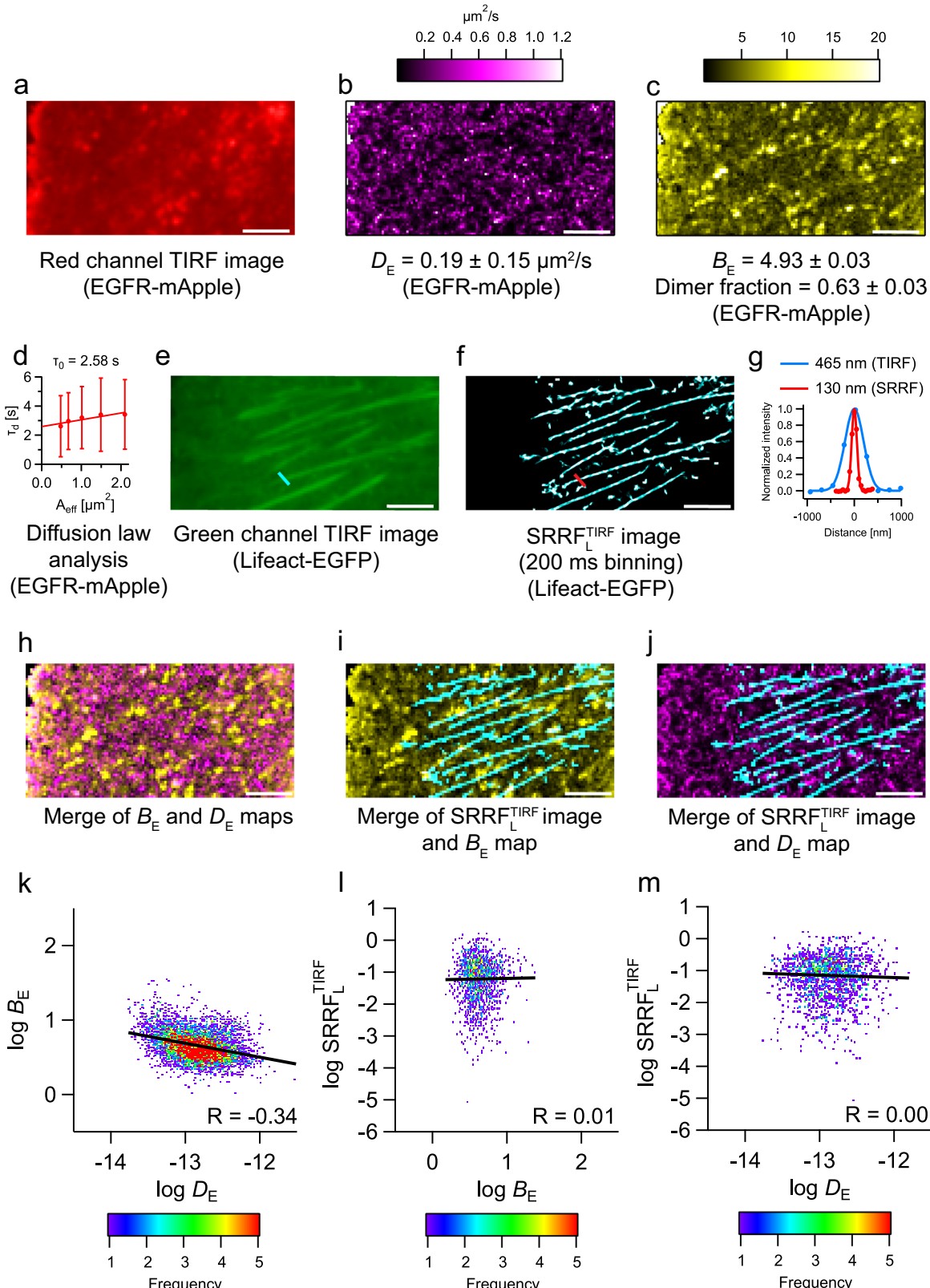

a

Red channel TIRF image
(EGFR-mApple)

b

$\mu m^2/s$

$D_E = 0.19 \pm 0.15\ \mu m^2/s$
(EGFR-mApple)

c

$B_E = 4.93 \pm 0.03$
Dimer fraction = 0.63 ± 0.03
(EGFR-mApple)

d

$\tau_0 = 2.58$ s

Diffusion law
analysis
(EGFR-mApple)

e

Green channel TIRF image
(Lifeact-EGFP)

f

$SRRF_L^{TIRF}$ image
(200 ms binning)
(Lifeact-EGFP)

g

— 465 nm (TIRF)
— 130 nm (SRRF)

h

Merge of $B_E$ and $D_E$ maps

i

Merge of $SRRF_L^{TIRF}$ image
and $B_E$ map

j

Merge of $SRRF_L^{TIRF}$ image
and $D_E$ map

k

R = -0.34

Frequency

l

R = 0.01

Frequency

m

R = 0.00

Frequency

To investigate the membrane organization, we used diffusion law analysis over different areas of CHO-K1 cells expressing EGFR-mApple. There were four kinds of areas that either include or exclude cytoskeletal structure or clusters. Independent of the presence or absence of cytoskeletal structures or clusters, EGFR shows positive intercepts in all cases, implying transiently

confined diffusion in lipid domains[40] (Supplementary Fig. 14, Supplementary Table 6). In this study, we use the term "lipid domains" to refer to 10–200 nm sized, dynamic, cholesterol and sphingolipid rich compartments in the cell membrane[41]. Furthermore, there is no statistically significant difference in intercepts obtained either for the whole-cell membrane, in

**Fig. 4 Multi-parametric analysis from a single dual-channel fluorescence data set. a** TIRF image of EGFR-mApple. **b** Diffusion map of EGFR-mApple ($D_E = 0.19 \pm 0.15$ μm$^2$/s). **c** Brightness map of EGFR-mApple ($B_E = 4.93 \pm 0.03$, dimer fraction-$m_e = 0.63 \pm 0.03$; mean ± SEM). **d** Diffusion law analysis of EGFR-mApple (refer Supplementary Fig. 16 for more details); intercept = 2.58 s; values are mean ± SD ($n = 6384$ pixels at 1 × 1 binning). **e** TIRF image of Lifeact-EGFP. **f** SRRF$_L^{TIRF}$ image (200 ms binning) of Lifeact-EGFP. **g** Normalized intensity profile across an actin fibre before and after SRRF microscopy. Thickness (FWHM of the Gaussian fit) of the actin fibre before SRRF (blue line in (**e**)) = 465 nm. The thickness of the actin fibre after SRRF (red line in (**f**)) = 130 nm. **h** Merge of $B_E$ and $D_E$ maps of EGFR-mApple. The $B_E$ pixels are coloured yellow, and the $D_E$ pixels are coloured magenta. Correlated pixels between the two maps are coloured white. **i** Merge of SRRF$_L^{TIRF}$ (Lifeact-EGFP) and $B_E$ (EGFR-mApple) maps. SRRF$_L^{TIRF}$ pixels are coloured cyan, and $B_E$ pixels are coloured yellow. Correlated pixels between the two maps are coloured white. The SRRF image was spatially binned to the same dimensions as the $B$ map. **j** Merge of SRRF$_L^{TIRF}$ (Lifeact-EGFP) and $D_E$ (EGFR-mApple) maps of EGFR-mApple. SRRF$_L^{TIRF}$ pixels are coloured cyan, and $D_E$ pixels are coloured magenta. Correlated pixels between the two maps are coloured white. The SRRF image was spatially binned to the same dimensions as the $D$ map. **k** 2D frequency plot of log $B_E$ vs log $D_E$ values from image (**h**). **l** 2D frequency plot of log SRRF$_L^{TIRF}$ vs log $B_E$ values from image (**i**). **m** 2D frequency plot of log SRRF$_L^{TIRF}$ vs log $D_E$ values from image (**j**). The scale bars in white measure 5 μm in images (**a**)–(**e**), (**h**)–(**j**). This figure uses cell 1 in Table 1. The reported values are averages obtained from analysis of an entire cell. The analyses were performed on six cells from three different batches of cells with similar results (Table 1). Source data is available as a Source data file.

membrane regions with or without clusters or with or without actin fibres (Supplementary Table 6, $p > 0.14$). We also obtain a negative correlation (Pearson's correlation coefficient-$R = -0.83$) between $D_E$ and the diffusion law intercept indicating that trapping is responsible for lower EGFR mobility as expected (Table 1).

Since all evaluations stem from the exact same data set we overlap the parameter maps and pairwise analyze their scatter plots to determine the interdependence of protein location with its dynamics. We performed a logarithmic transformation of the entire data set to identify power-law scaling between parameters (Fig. 4h–m). Using whole-cell data, we find a negative correlation between $B_E$ and $D_E$ ($R = -0.32$) (Table 1 and Fig. 4h, k), indicating that brighter particles diffuse slower than dimmer particles. This is unexpected since membrane diffusion is only weakly dependent on size[42]. Based on these observations, we hypothesize that either EGFR oligomers are located in a more viscous membrane environment, e.g., lipid domains[43] or clathrin-coated pits, or that we detect multiple EGFR molecules in larger complexes[39,44].

Brightness and SRRF do not show any correlation ($R = 0.00$, Table 1 and Fig. 4 i, l), implying that clustering is not directly linked to the cytoskeleton. Similarly, we do not see any correlation between EGFR diffusion and cytoskeletal structures ($R = 0.00$, Table 1 and Fig. 4j, m), unlike in the case of Lifeact. This also verifies our earlier observation that diffusion law intercepts are independent of the presence or absence of actin. Similar results were obtained by performing the experiment using a sCMOS camera as detector (Supplementary Figs. 15 and 16).

**Modulation of EGFR dynamics and clustering.** Next, we investigated the effect of the cognate ligand EGF (epidermal growth factor), the lipid microenvironment and the actin cytoskeleton on the dynamics and clustering of EGFR. For the latter, we used drugs to depolymerize the actin cytoskeleton (Latrunculin A—LAT-A) or extract cholesterol from the cell membrane (methyl-β-cyclodextrin—MβCD, and cholesterol oxidase—COase).

Upon addition of 100 ng/ml of EGF (Supplementary Fig. 17), $D_E$ reduced 3.8 fold, EGFR clusters appeared, and N&B analysis showed the formation of at least trimers (Table 2, Supplementary Table 7). Concomitantly, the FCS diffusion law intercept increased 3.4 fold. The decrease in $D_E$ and increase in $B_E$ and FCS diffusion law intercept, all indicate the dynamic formation of EGFR clusters.

The TIRF images of cells treated by LAT-A to disrupt the cytoskeleton showed homogeneous fluorescence inside the cells due to absence of fibres. At the same time, $D_E$ decreases by a

factor of 2.9 (Supplementary Fig. 18). The FCS diffusion law intercept shows a 1.9 fold increase as corralling by the cytoskeleton is released[45]. And finally, molecular brightness remained similar to the resting state, indicating that the EGFR oligomerization state is not disturbed.

Upon modulation of the cholesterol concentration by the addition of 3 mM MβCD (Supplementary Fig. 19) or 1 U/ml COase (Supplementary Fig. 20), a reduction in $D_E$, increase in diffusion law intercept and an increase in $B_E$ is observed, with COase having a stronger effect on all three measured parameters. The increase of diffusion law intercept and $B_E$ and decrease of $D_E$ implies the dynamic formation of EGFR clusters. Cholesterol removal leads to receptors being spatially less constrained and hence leads to an increase in the probability of cluster formation[46].

## Discussion

The combination of fast acquisition with small pixels and GPU-based data treatment allows the simultaneous determination of super-resolved structure and millisecond molecular dynamics from the exact same pixels over a whole image without any special sample preparation and almost in real-time. This provides a tool to determine correlations between molecular structure and dynamics using a single, or better yet, two colours as illustrated by Lifeact and EGFR. It is important to note that, although we have used SRRF, SOFI or deconvolution methods simultaneous multiparametric spatiotemporal microscopy is also possible with other computational super-resolution tools, e.g., 3B analysis[47] or SPARCOM[48], since it does not need any special acquisition modalities but can be applied to the same data as the spectroscopic techniques measuring dynamics. The mutual consistency of the computational super-resolution method and the dynamics method provides a better estimate of dynamics parameters and allows corrections of possible structural artefacts.

We demonstrate the usefulness of the mutual consistency of dynamics and structural data on the example of SRRF. The resolution of SRRF was determined at actin branch points in the meshwork. The FRC and P2P were 90–136 nm and 136–240 nm, respectively, similar to a previously published FRC value of 108 nm[49]. However, SRRF images, similar to other computational microscopy and super-resolution techniques, are prone to show artefacts and the user has to choose parameters carefully to optimize the size of structural features and resolution but simultaneously limit the extent of artefacts. We, therefore, devised a correction strategy for the reduction of these artefacts to improve the consistency between dynamic and structural data. For this purpose, we first filtered Imaging FCS data by retaining only those pixels that can be attributed to structures in the TIRF image.

**Table 1 Multi-parametric analyses of CHO-K1 cells labelled with Lifeact-EGFP and EGFR-mApple.**

| Cells | Number of Pixels | $D_E$ (µm²/s) | $B_E$ a | Fraction of molecules in dimer $(m_e)$ a | R (log $B_E$, log $D_E$) | R (log SRRF_TIRF, log $B_E$) | R (log SRRF_TIRF, log $D_E$) | FWHM of Gaussian fits to normalized intensity profiles from SRRF b (nm) | Diffusion law intercept (s) |
|---|---|---|---|---|---|---|---|---|---|
| 1 | 6690 | 0.19 ± 0.15 | 4.93 ± 0.03 | 0.63 ± 0.03 | −0.34 | 0.01 | 0.00 | 157 ± 26 | 2.58 |
| 2 | 3811 | 0.21 ± 0.19 | 5.28 ± 0.03 | 0.80 ± 0.03 | −0.19 | −0.00 | −0.07 | 115 ± 18 | 2.20 |
| 3 | 6720 | 0.31 ± 0.31 | 4.97 ± 0.03 | 0.65 ± 0.03 | −0.40 | 0.07 | −0.07 | 140 ± 24 | 0.76 |
| 4 | 6061 | 0.11 ± 0.09 | 4.78 ± 0.03 | 0.56 ± 0.03 | −0.50 | −0.05 | 0.04 | 135 ± 20 | 5.16 |
| 5 | 5119 | 0.26 ± 0.24 | 4.03 ± 0.03 | 0.18 ± 0.02 | −0.22 | −0.05 | 0.05 | 143 ± 19 | 1.59 |
| 6 | 3534 | 0.27 ± 0.25 | 4.95 ± 0.06 | 0.64 ± 0.03 | −0.25 | 0.04 | 0.01 | 103 ± 9 | 1.65 |
| Mean | | **0.23 ± 0.22** | **4.82 ± 0.04** | **0.58 ± 0.03** | **−0.32 ± 0.12** | **0.00 ± 0.05** | **0.00 ± 0.05** | **132 ± 27** | **2.32 ± 1.52** |

a The reported values are mean ± SEM. The SEM was calculated using Eq. 19 in the supplement. In all other cases, the values are mean ± SD.
b The reported values are mean ± SD from three positions in each cell measurement.
Mean values are represented in bold.

**Table 2 Multi-parametric analyses of CHO-K1 cells labelled with Lifeact-EGFP and EGFR-mApple subjected to various drug treatments.**

| | Cell | $D_E$ (µm²/s) | $B_E$ a | Min order oligo | Fraction of molecules in oligomer $(e_{oligomer})$ a | R (log $B_E$, log $D_E$) | R (log SRRF_TIRF, log $B_E$) | R (log SRRF_TIRF, log $D_E$) | Diffusion law intercept (s) |
|---|---|---|---|---|---|---|---|---|---|
| 100 ng/ml EGF | 1 | 0.05 ± 0.05 | 6.52 ± 0.13 | 3 | 0.71 ± 0.03 | −0.25 | −0.05 | −0.03 | 7.26 |
| | 2 | 0.04 ± 0.06 | 5.87 ± 0.13 | 3 | 0.55 ± 0.03 | −0.27 | −0.01 | 0.01 | 8.03 |
| | 3 | 0.08 ± 0.05 | 9.97 ± 0.22 | 5 | 0.79 ± 0.03 | −0.45 | −0.1 | 0.08 | 8.20 |
| | Mean | **0.06 ± 0.06** | **7.45 ± 0.12** | | | **−0.32 ± 0.11** | **−0.05 ± 0.05** | **0.02 ± 0.06** | **7.83 ± 0.50** |
| 3 µM LAT-A | 1 | 0.06 ± 0.04 | 5.13 ± 0.05 | 2 | 0.73 ± 0.03 | −0.47 | NA | NA | 5.59 |
| | 2 | 0.11 ± 0.10 | 4.39 ± 0.03 | 2 | 0.36 ± 0.02 | −0.19 | NA | NA | 2.81 |
| | 3 | 0.07 ± 0.02 | 6.00 ± 0.04 | 3 | 0.58 ± 0.02 | −0.52 | NA | NA | 4.94 |
| | Mean | **0.08 ± 0.06** | **5.17 ± 0.03** | | | **−0.39 ± 0.18** | **NA** | **NA** | **4.45 ± 1.45** |
| 3 mM MβCD | 1 | 0.03 ± 0.03 | 7.67 ± 0.13 | 3 | 0.99 ± 0.04 | −0.23 | −0.02 | 0.00 | 10.59 |
| | 2 | 0.04 ± 0.05 | 8.03 ± 0.18 | 4 | 0.73 ± 0.03 | −0.25 | 0.01 | −0.04 | 9.97 |
| | 3 | 0.07 ± 0.06 | 4.51 ± 0.06 | 2 | 0.42 ± 0.04 | −0.16 | 0.05 | −0.06 | 8.32 |
| | Mean | **0.05 ± 0.04** | **6.74 ± 0.10** | | | **−0.21 ± 0.05** | **0.01 ± 0.04** | **−0.03 ± 0.03** | **9.63 ± 1.17** |
| 1 U/ml COase | 1 | 0.03 ± 0.04 | 16.33 ± 0.07 | 8 | 0.90 ± 0.02 | −0.22 | −0.04 | −0.05 | 24.07 |
| | 2 | 0.03 ± 0.03 | 5.78 ± 0.03 | 3 | 0.53 ± 0.02 | −0.34 | −0.07 | 0.02 | 20.11 |
| | 3 | 0.03 ± 0.05 | 9.33 ± 0.05 | 4 | 0.94 ± 0.02 | −0.37 | 0.09 | 0.18 | 27.6 |
| | Mean | **0.03 ± 0.04** | **10.48 ± 0.06** | | | **−0.31 ± 0.08** | **−0.01 ± 0.09** | **0.05 ± 0.12** | **23.93 ± 3.75** |

a The reported values are mean ± SEM. The SEM was calculated using Eq. 19 in the supplement. In all other cases, the values are mean ± SD.
Mean values of the 3 cells for each treatment are represented in bold.

However, this does not remove artefacts due to slow-moving bright aggregates, as identified by FCS, and which are not actual structural features. By using only those pixels that contain $D$ consistent with diffusion on fibres we can correct the SRRF image and remove artefacts that are created by mobile bright complexes. This strategy significantly improves the accuracy and precision of $D$ and removes artefacts in the super-resolved images. Even more importantly, the correlation between fibre position of SRRF and probe dynamics of Imaging FCS provide mutual support for the two techniques. The SRRF position is corroborated by the lower COV of FCS, and vice versa. This allows a clearer distinction between diffusion on and off fibres. The slower $D_{L,on(4)}^{TIRF,D} = 1.47 \pm 0.99$ μm²/s implies binding and unbinding of Lifeact and thus a slowdown in mobility, while the faster $D_{L,off(4)}^{TIRF} = 4.02 \pm 3.98$ μm²/s indicates free diffusion of Lifeact or Lifeact bound to G-actin[26]. It is important to note that this thresholding-based TIRF masking followed by $D$ filtering strategy to correct for SRRF artefacts can be applied only on samples where a distinction can be made between the different regions of the cell. Empirically, we find that, the SNR must be at least 3 for efficient filtering using this approach.

In the case of two-colour measurements, in order to estimate the oligomerization of EGFR in the resting state, we first estimated the proportion of fluorescent molecules of mApple. We found that 55 ± 1% of mApple are fluorescent which is well within the upper range of literature values for red FPs of 20–70%[32]. Using this calibration, we find that 58 ± 3% of transfected EGFR molecules exist as dimers in CHO-K1 cells, which do not have endogenous expression of EGFR[38]. The existence of preformed oligomers is consistent with literature with up to ~65% dimerization previously detected[39,50,51] in the same cell line.

FCS diffusion law analysis showed that EGFR has a positive intercept indicating that it partitions into domains. We observed that brighter particles diffuse slower compared to dimmer particles, whereas there was no correlation between the diffusion of EGFR and the underlying cytoskeleton. This indicates that the domain partitioning of EGFR is primarily determined by EGFR-membrane interactions, possibly sub-resolution clustering and inter-EGFR interactions but independent of EGFR-actin interactions. As it has been shown previously that EGFR dynamics changes with cytoskeleton disruption[40], our observations suggest that it is not the static cytoskeleton itself that changes EGFR dynamics but that the influence is indirect and cytoskeleton coupling to the plasma membrane and resulting membrane changes are responsible for changes in EGFR dynamics[52].

Further investigation of the influence of ligand binding (EGF) or the change of the microenvironment by modulating cholesterol content (MβCD or COase), shows that in all cases there is not only a decrease in $D$ but also an increase in $B$ and the diffusion law intercept indicating receptor clustering. In all treatments we see the same negative correlation of $D$ and $B$ as well as of $D$ and the diffusion law intercept. Furthermore, the treatments show that the diffusion law intercept is proportional to the brightness, reiterating that $B$ and $D$ exhibit a negative correlation. This suggests that even after treatment the EGFR clusters are still located in cellular regions of decreased molecular mobility such as lipid domains or clathrin-coated pits.

Independent of the treatment, the correlation between SRRF and $B$ and SRRF and $D$ is close to zero. These measurements suggest that actin does not have a direct influence on EGFR diffusion or oligomerization at length scales of ~700 nm, i.e., the linear dimension of the observation area of one pixel.

On the contrary, cytoskeleton disruption by LAT-A leads to a decrease in $D$ and an increase in the diffusion law intercept. This indicates that actin has an effect on EGFR diffusion. These observations corroborate the hypothesis that the static cytoskeleton does not change EGFR dynamics or organization directly, but that the influence of the cytoskeleton is indirect. Changes to cytoskeleton organization and its interaction with the cell membrane lead to altered membrane properties which in turn result in changes in EGFR dynamics or clustering.

LAT-A treatment on average did not show an increase in oligomerization while EGF led to the formation of visible clusters, which is consistent with literature[40,53]. The emerging picture suggests that EGFR exists as monomers and dimers at least partially in cholesterol-dependent domains and it can be freed from these domains and cluster upon cholesterol removal. The actin cytoskeleton does not directly influence EGFR but its main effect is on the organization of the cell membrane which in turn influences EGFR dynamics and organization.

Finally, on a technical note, sCMOS cameras can be used as alternative detectors to EMCCDs although there are some differences. EMCCD cameras have somewhat better SNR at low light levels and are thus better detectors under these circumstances. However, they have also much larger pixels which is counterproductive when spatial super-resolution is to be achieved, unless corrected by changing the magnification. sCMOS cameras on the other hand can be read-out at least one order of magnitude faster than EMCCDs and thus can provide access to faster diffusing particles, but in that case, need higher laser power to obtain sufficient signal[54,55]. Despite these differences, both detectors provide the same absolute parameter values in our measurements albeit with different SNR (EMCCD-6.3, sCMOS-1.7).

Living systems are highly dynamic with processes happening on spatial scales well below the optical diffraction limit and on time scales on the millisecond scale or faster. Therefore, the extraction of the maximum information available on biological processes requires the simultaneous acquisition of data with high spatiotemporal resolution. This poses particular problems to data recording and data evaluation strategies that allow data treatment in an acceptable time frame, ideally in real-time. We used GPU-based data evaluation and modern camera technology to optimize data evaluation of fluorescence microscopy images by selected spatial and temporal binning to extract multiple physicochemical parameters in parallel from a single data set, almost in real time, using standard and widely available instrumentation. The simultaneous analysis of structural and dynamics data allow categorizing dynamic data according to underlying structures and removing artefacts from computational microscopy and super-resolution images. And in multi-parametric evaluations, parameter correlations and the lack thereof provide information not available on sequentially acquired data due to the dynamical nature of biological samples as shown in the example of EGFR. This approach is easily extendable to other fluorescence parameters, does not require specialized instrumentation, and thus is immediately applicable to a wide range of situations. We have shown that on a TIRF microscope but the same strategy is applicable to any illumination system that can efficiently capture a cross section of a 3D sample. Especially the lack of requirement of any customized instrumentation and the provision of the necessary evaluation software makes this approach immediately applicable to a wide set of researchers.

## Methods

**Plasmid construction.** The N-terminal of the mApple sequence was coupled with the first 45 nucleotides of the $RP2$ gene to target it to the plasma membrane[34]. The plasmid containing the pRK-5 vector backbone with a pUC origin of replication, cytomegalovirus (CMV) promoter, SV40 polyadenylation signal site, ampicillin resistance gene cloned with the membrane-targeted mApple was obtained from VectorBuilder Inc. (Illinois, USA). We refer to this plasmid as plasma membrane-targeted mApple (PMT-mApple). To create a double labelled PMT-mApple₂ the PMT-mApple plasmid was digested with NheI and HindIII to create the backbone. The plasmid was also separately digested with SpeI and HindIII to get the mApple sequence insert. The backbone and the insert were ligated (this was possible

because SpeI and NheI are isocaudomers) using T4 DNA ligase (M0202S; New England BioLabs, Massachusetts, USA) to create PMT-mApple$_2$. The EGFR-mApple plasmid was created by replacing the PMT sequence with human EGFR sequence using restriction digestion with AgeI and SpeI and ligation using T4 DNA ligase. The plasmid maps created using SnapGene® software (version 4.3.11, GSL Biotech LLC, Illinois, USA) are shown in Supplementary Fig. 21.

**Cell culture**. A detailed protocol describing the steps in preparation of live-cell samples is provided in Protocol Exchange[56]. CHO-K1 (Chinese Hamster Ovary) cells (CCL-61) were obtained from ATCC (Manassas, Virginia, USA) and were cultured in Dulbecco's Modified Eagle Medium (DMEM/High glucose with L-glutamine, without sodium pyruvate – #SH30022.FS; HyClone, GE Healthcare Life Sciences, Utah, USA) supplemented with 1% penicillin-streptomycin (#15070063, Gibco, Thermo Fisher Scientific, Massachusetts, USA), and 10% foetal bovine serum (FBS; #10270106, Gibco, Thermo Fisher Scientific, Massachusetts, USA) at 37 °C in a 5% (v/v) CO$_2$ environment (Forma Steri-Cycle CO$_2$ incubator, Thermo Fisher Scientific, Massachusetts, USA).

Lifeact-EGFP plasmid (Addgene plasmid #58470) was a gift from Prof. Wu Min (NUS, Singapore). All the plasmids were amplified using ZymoPURE II Plasmid Midiprep Kit (D4201; Zymo Research, California, USA) and their concentration and purity were confirmed by a UV–Vis spectrophotometer (NanoDrop 2000, Thermo Fisher Scientific, Massachusetts, USA).

For transfection, cell cultures that were ~90% confluent were used. The spent media removed from the culture flask was discarded. The flask was washed twice with 5 ml PBS (phosphate-buffered saline; without Ca$^{2+}$ and Mg$^{2+}$). 2 ml Trypsin-EDTA (0.5%; #15400054, Gibco, Thermo Fisher Scientific, Massachusetts, USA) was added and the flask was incubated at 37 °C for 2–3 min to detach the cells. 5 ml culture media was added to the flask to inhibit trypsin. The media containing the detached cells was centrifuged at 200 × g for 3 min. The supernatant was discarded and the cell pellet was resuspended in 5 ml PBS. Cell counting was done using a cell counter (Bio-Rad, Singapore). The required number of cells was centrifuged (#5810, Eppendorf, Hamburg, Germany) at 200 × g for 3 min. The supernatant was discarded and the cells were resuspended in R buffer (Neon Transfection Kit, Thermo Fisher Scientific, Massachusetts, USA). Suitable amounts of Lifeact-EGFP and EGFR-mApple (or PMT-mApple, or PMT-mApple$_2$) (100 ng for Lifeact-EGFP, PMT-mApple, PMT-mApple$_2$; 1 μg for EGFR-mApple) plasmids were mixed with the cells for co-transfection. The cells were electroporated using Neon Transfection System (Thermo Fisher Scientific, Massachusetts, USA) according to the manufacturer's protocol (electroporation settings: pulse voltage = 1,000 V, pulse width = 30 ms, and pulse no. = 2). After transfection, the cells were seeded onto culture dishes (#P35G-1.5-20-C, MatTek, Massachusetts, USA) containing DMEM (supplemented with FBS; no antibiotic). The cells were incubated at 37 °C and 5% CO$_2$ for 36–48 h before measurements.

Before EGFR measurements, the cells were washed with HBSS (Hank's Balanced Salt Solution, with Ca$^{2+}$ and Mg$^{2+}$; #14025134; Gibco, Thermo Fisher Scientific, Massachusetts, USA) and starved in DMEM not containing phenol red (#21063029; Gibco, Thermo Fisher Scientific, Massachusetts, USA) for at least 4 h. To avoid internalization of EGFR, internalization inhibitors were added to the cells 30 min before the measurements. The internalization inhibitors used were 2 mM NaF, 10 mM NaN$_3$ and 5 mM 2-deoxy-D-glucose (Sigma-Aldrich, Singapore).

For the drug treatments, the cells were first prepared as described in the previous paragraph, and then treated with different drugs. Working concentrations of the drugs were prepared in DMEM not containing phenol red. For EGF treatment, the cells were stimulated with 100 ng/ml of human epidermal growth factor (hEGF; E9644, Sigma-Aldrich, Singapore) for 20 min. Actin cytoskeleton depolymerisation was done by treating the cells with 3 μM LAT-A (L5163, Sigma-Aldrich, Singapore) for 15 min. Cholesterol depletion was done by treating the cells for 30 min with 3 mM MβCD (C4555, Sigma-Aldrich, Singapore) or 1 U/ml COase (C8649, Sigma-Aldrich, Singapore).

**Supported lipid bilayer**. All glassware (slides, coplin jars and round-bottom flasks) were cleaned thoroughly using an alkaline cleaning solution (Hellmanex III, Hellma Analytics, Müllheim, Germany) diluted ten times. They were then submerged in the cleaning solution and sonicated (Elmasonic S30H, Elma Schmidbauer GmbH, Singen, Germany) for 30 min. This was followed by washing with ultrapure water (Milli-Q, Merck, New Jersey, USA) and another round of sonication for 30 min after submerging in 2 M H$_2$SO$_4$. The glassware was then washed and submerged in ultrapure water for the third round of sonication for 30 min. After air-drying, the glassware was used for SLB preparation. An O-ring mould was filled with a silicone elastomer (SYLGARD 184 Silicone Elastomer Kit, Dow, Michigan, USA) and cured at 65 °C overnight. The O-rings were then removed carefully using forceps, attached to a slide using the silicone elastomer and cured for 3 h at 65 °C. Unused glassware and O-rings were stored in 100% ethanol for later use.

0.5 mM DOPC(Avanti Polar Lipids, Alabama, USA) and 50 nM Rhodamine PE (14:0 Liss Rhod PE, Avanti Polar Lipids, Alabama, USA)/100 nM Lipilight 488 (Idylle, Paris, France) were mixed in a round-bottom flask. The solution was evaporated in a rotary evaporator (Rotavapor R-210, Büchi, Flawil, Switzerland) and a thin lipid film was left behind. The lipid film was dissolved in 2 ml of a buffer solution (10 mM HEPES, 150 mM NaCl, pH 7.4) and sonicated until the solution

became clear. 200 μl of the lipid solution was pipetted into an O-ring attached to a slide. The slide was incubated at 65 °C for 1 h to allow the SLB to form by vesicle fusion. Then it was cooled at room temperature (25 °C) for 30 min. The SLB was washed multiple times with the buffer solution (by removing 100 μl of lipid solution inside the O-ring and adding 100 μl of buffer) to get rid of excess, unfused vesicles and then used for measurements.

**Instrumentation**. The TIRF microscopy set-up included an inverted epi-fluorescence microscope (IX83, Olympus, Japan), a motorized TIRF illumination combiner (cellTIRF-4Line IX3-MITICO, Olympus, Japan), and a dual-emission image splitter (OptoSplit II; Cairn Research, Faversham, UK). We used either an electron-multiplying charge-coupled device (EMCCD; iXon$^{EM}$ + 860, 24 μm pixel size, 128 × 128 pixels, Andor, Oxford Instruments, UK) camera or a scientific complementary metal oxide semiconductor (sCMOS; Sona 4.2B-11, 11 μm pixel size, 2048 × 2048 pixels, Andor, Oxford Instruments, UK) for detection. 488 nm (LAS/488/100, Olympus, Japan) and 561 nm (LAS/561/100, Olympus, Japan) lasers were connected to the TIRF illumination combiner. We used a 100×, NA 1.49 oil-immersion objective (Apo N, Olympus, Japan) and a magnification changer slider (IX3-cAS, Olympus, Japan) to increase magnification two-fold to 200× where required. For the cell measurements, 37 °C temperature and 5% CO$_2$ atmosphere were maintained using an on-stage incubator (Chamlide TC, Live Cell Instrument, South Korea). The laser power used was 100 μW for the 488 nm laser and 900 μW (EMCCD) for the 561 nm laser (as measured at the back aperture of the objective).

For the dual-channel measurements, the fluorescence light was passed through a dichroic (ZT 405/488/561/640rpc, Chroma Technology Corp, Vermont, USA) and emission filter (ZET405/488/561/640 m, Chroma Technology Corp, Vermont, USA), and split by the image splitter on two halves of the camera chip. The image splitter was fitted with an emission dichroic (FF560-FDi01; Semrock, New York, USA) and band-pass filters (510AF23 and 585ALP, respectively; Omega Optical, Vermont, USA). A bright-field image of a stage micrometre was used to align the image splitter. This was done in μManager (version 1.4.14, https://micro-manager.org). The image was aligned in both the channels following the manual instructions. To check how good the alignment was, a self-written program in μManager was used to find the similarity in both channels. We considered the channels sufficiently aligned if the similarity was ≥95%. In the case of EMCCD, the measurements were done by recording a stack of 50,000 frames of 128 × 128 pixels at 500 frames per second (fps) (for cell measurements)/1,000 fps (for bilayer measurements). Andor Solis (version 4.31.30037.0-64-bit) was used for image acquisition. The kinetic mode of image acquisition was used and the 'baseline clamp' was always used to minimize the baseline fluctuation. The camera was operated using 10 MHz pixel readout speed. The maximum analog-to-digital gain was set to 4.7 and 0.45 μs vertical shift speed was used. The EM gain used was 300.

**Calibration of PSF for Imaging FCS**. To calibrate the PSF for the different experimental setups, the relevant bilayer (Rhodamine PE for 561 nm and Lipilight 488 for 488 nm) measurement file of 20 × 20 pixels is loaded in the ImFCS plugin in ImageJ. The values used for the parameters were: frame time = 0.001 s, correlator $(p, q)$ = (16, 9), pixel size = 24 μm (EMCCD)/11 μm (sCMOS), NA = 1.49, $\lambda_1$ = 506 nm (green channel)/565 nm (red channel). The program computed and displayed a plot of the $D$ values for various combinations of PSF and binning values.

**Data analyses**. The data analyses were performed on a computer with the following configuration—Windows 10 Home 64-bit operating system, Intel® Core™ i7-7800X CPU @ 3.50 GHz processor, 32 GB RAM, NVIDIA TITAN Xp GPU with 3840 CUDA cores and 12.3 GB memory. A step-by-step protocol for various data analyses is provided in Protocol Exchange[57].

*FCS*. The image stacks from the cell measurements were loaded in Imaging FCS[58] 1.52 plugin for Fiji and the ACFs calculated. The source code is available here[59]. The values used for the parameters were: frame time = 0.002 s, correlator (p, q) = (16, 12), pixel size = 24 μm, NA = 1.49, $\lambda_1$ = 565 nm (red channel). The EMCCD data analysis was performed at 1 × 1 binning for ×100 magnification. Bleach correction[27] was performed with a polynomial of order 8. ACFs were fitted with a one-component diffusion model[60].

$$G(\tau) = \frac{1}{N}\frac{g(\tau)}{g(0)} + G_\infty \qquad (1)$$

$$g(\tau) = \left( \frac{\sqrt{4D\tau + \omega_{xy}^2}}{a\sqrt{\pi}} \left( e^{-\frac{a^2}{4D\tau + \omega_{xy}^2}} - 1 \right) + erf\left( \frac{a}{\sqrt{4D\tau + \omega_{xy}^2}} \right) \right)^2 \qquad (2)$$

where $a$ is the pixel size, $\tau$ is the lag time, $N$ is the number of particles, $D$ is the diffusion coefficient, $\omega_{xy}$ is the PSF (xy) ($1/e^2$ radius) direction. To exclude outliers or non-converged fits, only data with $0.01 < D < 10$ μm$^2$/s were considered for further analysis. The PSF was estimated using the method described here[29].

*FCS diffusion law*. The image stack was analysed using the Imaging FCS 1.52 plugin for Fiji. The values used for the parameters were: frame time = 0.002 s, correlator

**Table 3 Acquisition parameters for the various experimental configurations.**

| Camera | Analysis | Laser wavelength (nm) | Time per frame (ms)[a] | No. of frames | Image pixel size (nm) | Map dimensions (Pixels) |
|---|---|---|---|---|---|---|
| EMCCD | FCS | 488 | 2 | 50,000 | 240 | 46 × 46 |
| Fig. 2 | SRRF | 488 | 200 | 500 | 24 | 460 × 460[b] |
| EMCCD | FCS | 561 | 2 | 50,000 | 240 | 120 × 56 |
| Fig. 4 | Diffusion law | 561 | 2 | 50,000 | 240 | 114 × 56 |
| | N&B | 561 | 20 | 5,000 | 240 | 120 × 56 |
| | SRRF | 488 | 200 | 500 | 48 | 600 × 280[c] |

[a]Acquisition time was 2 ms; longer times are reached by time binning of frames.
[b,c]The original image has a pixel size of 120 nm (b) and 240 nm (c) at 1 × 1 binning, respectively. During the SRRF analysis, virtual sub-pixels of sizes 24 nm (b) and 48 nm (c), respectively, are created.

(p, q) = (16, 12), pixel size = 24 μm (EMCCD), magnification = 100, NA = 1.49, $\lambda_1$ = 565 nm (red channel), PSF (xy) = 0.96 (EMCCD red channel) and bleach correction = polynomial of order 8. In the "Diff. Law" tab, the diffusion law plot was generated and fit with a straight line for square binning 1–5 (EMCCD).

*N&B.* The 2 ms, 50,000 frames image stack was temporally binned (sum binning of 10 frames each) to a 20 ms, 5000 frames stack in Fiji using the Image → Transform → Bin command sequence. The images were analysed using the Imaging FCS 1.52 plugin. The values used for the parameters were: frame time = 0.02 s, binning = 1 × 1 (EMCCD), and bleach correction = polynomial of order 8. An intensity filter with a suitable range was set, as defined by the background intensity in an image, to exclude the background pixels and include only the pixels containing the cell. A dark image (image taken with the camera shutter closed) with the same spatiotemporal dimensions as the measurement image was loaded for background correction in the "Bgr NUM" tab. Then in the "N&B" tab, "G1" was selected in the "NB mode" and the "N&B" button in "N&B analysis" was pressed to generate the N&B maps. The N&B equations are defined as[61]:

$$N = \frac{(\langle I \rangle - \text{offset})^2}{\sigma^2 - \sigma_0^2} \quad (3)$$

$$B = \frac{\sigma^2 - \sigma_0^2}{\langle I \rangle - \text{offset}} \quad (4)$$

where $N$ is the apparent number of particles in the observation volume, $\langle I \rangle$ is the average intensity, offset is the intensity offset of EMCCD, $\sigma^2$ is the variance of the signal, $B$ is the apparent brightness of a particle, and $\sigma_0^2$ is the variance of the readout noise in the EMCCD. The offset and $\sigma_0^2$ can be obtained from a dark image captured by the EMCCD. The variance is typically affected by shot noise, and hence we used the covariance. This is referred to as G1 analysis.

$$\text{Covariance}(F(t), F(t + \Delta t)) = \langle F(t)F(t + \Delta t) \rangle \quad (5)$$

where $F(t)$ is the fluorescence signal at time $t$ and $F(t + \Delta t)$ is the fluorescence signal at time $t + \Delta t$.

The fraction of EGFR molecules present as dimers ($m_e$) is determined using the equation below:

$$m_e = \frac{r - 1}{p} \quad (6)$$

where $r$ is the ratio of the brightness of EGFR to the brightness of monomer, $p$ is the proportion of the molecules which are fluorescent. The procedure to determine $p$ based on the brightness of monomer and dimer is provided in supplement. The error propagation to estimate the error associated with the dimer fraction is also provided in Supplementary Note 3.

*SRRF.* The 2 ms, 50,000 frames image stack was temporally binned (average binning of 100 frames each) to a 200 ms, 500 frames stack (for Supplementary Fig. 4 other temporal binnings were also used) in Fiji using the Image → Transform → Bin command sequence. The NanoJ-SRRF[24] plugin (version 1.14Stable1) was used. The "SRRF analysis" option in the plugin was chosen. The default settings were used except for using Temporal Radiality Auto-Correlations (TRAC) of order 2 for the temporal analysis. Ring radius = 0.5, radiality magnification = 5, axes in ring = 6. The "display_mode" used was "radiality".

To determine the fibre thickness, the straight line tool in Fiji was used to draw a line segment across a fibre in the generated SRRF images. The "Plot Profile" function was used to generate an intensity histogram, which was fitted using the "Curve Fitting" tool in Fiji. The curve fitting function used was a Gaussian given by the equation below.

$$y = a + (b - a)e^{\frac{-(x-c)^2}{2d^2}} \quad (7)$$

where $a$ is the offset, $b - a$ is the height at the centre $c$, and $d$ is the standard deviation of the Gaussian. The SRRF images had black borders so the image was cropped to get rid of them. As a result, the original acquired image was cropped before performing FCS, diffusion law and N&B analyses to maintain the same cell area in all the maps (for the image dimensions, refer Table 3).

*SOFI.* The 200 ms temporally binned stack as described in the SRRF section above was used. In the NanoJ-SRRF plugin, the "SRRF analysis" option in the plugin was chosen. The ring radius was set to 3, radiality magnification = 1, axes in ring = 2.

*Deconvolution.* The 2 ms, 50,000 frames image stack was temporally binned (average binning of all the frames) to obtain a single time-averaged image in Fiji. This image was deconvolved with Huygens Professional (version 20.04, Scientific Volume Imaging, The Netherlands). The microscope type was set to confocal and the Z sampling interval was set to the ideal Nyquist value of 100 nm. The X and Y sampling intervals were set as 120 nm. The optical parameters used were: NA = 1.49, refractive indices of lens immersion oil and embedding medium of 1.518 and 1.38[62], respectively, objective quality = good. The channel parameters used were: backprojected pinhole = 2,500 nm, excitation wavelength = 488 nm, emission wavelength = 510 nm, excitation fill factor = 2. An automatically generated theoretical PSF was used. The background value of the image was automatically estimated with the emission mode set to lowest and the area radius set to 0.7 microns. The classic maximum likelihood estimation (CMLE) was used as the deconvolution algorithm with maximum iterations = 50, SNR = 100, quality threshold = 0.05, iteration mode = optimized, brick layout = auto.

**SNR calculation**. For the SNR calculation on the EMCCD and sCMOS, an area of 20 × 20 pixels was chosen inside and outside a cell. The area outside the cell was used to estimate the background for SNR calculation using the equation below.

$$\text{SNR} = \frac{\langle I \rangle - \langle I_{\text{bgr}} \rangle}{\sigma} \quad (8)$$

where $\langle I \rangle$ is the average intensity inside the cell (signal), $\langle I_{\text{bgr}} \rangle$ is the average intensity outside the cell (background), and $\sigma$ is the standard deviation of the signal inside the cell.

For the SNR calculation between fibre and non-fibre areas, an area of 3 × 3 pixels was chosen on and off-fibres. The SNR was calculated using the equation below.

$$\text{SNR} = \frac{\langle I_{\text{fibre}} \rangle - \langle I_{\text{off-fibre}} \rangle}{\sigma_{\text{fibre}}} \quad (9)$$

where $\langle I_{\text{fibre}} \rangle$ is the average intensity on the fibre, $\langle I_{\text{off-fibre}} \rangle$ is the average intensity off fibre, and $\sigma_{\text{fibre}}$ is the standard deviation of the signal on the fibre.

**Statistical analysis**. The frequency plots in Fig. 4 were generated using the "Bivariate histogram" command in IgorPro© (Wavemetrics Inc., Oregon, USA). A linear fit was performed to estimate the correlation coefficient ($R$). Graphpad Quickcalcs was used to perform $t$-tests. Only those pixels were included in the scatter plots that had valid values for both parameters.

**Reporting summary**. Further information on research design is available in the Nature Research Reporting Summary linked to this article.

## Data availability
The data sets generated during the current study are available from the corresponding author on reasonable request. Source data are provided with this paper.

## Code availability
The Imaging FCS 1.52. ImageJ plugin is available at https://github.com/ImagingFCS/Imaging_FCS_1_52.[59] or alternatively is included in the ImageJ update site.

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

## Acknowledgements

T.W. gratefully acknowledges funding from the Singapore Ministry of education (MOE2016-T3-1-005 and MOE2016-T2-2-121). H.B., W.H.T., and X.W.N. are recipients of research scholarships of the National University of Singapore. The authors would like to acknowledge Angela Koh Phei San for assistance in plasmid design and construction, Daniel Aik Ying Kia for assistance in lipid bilayer preparation, and Ashwin Venkata Subba Nelanuthala for assistance in derivation of formulae used in this study. We gratefully acknowledge the support of NVIDIA Corporation with the donation of the TITAN Xp GPU used for this research.

## Author contributions

T.W. conceived and designed the study. J.S. and W.H.T. wrote programs for GPU analysis. H.B. and X.W.N. performed the experiments. J.S. and H.B. analyzed the data. J.S., H.B., W.H.T. and T.W. wrote the manuscript. A.R. and T.W. supervised the study.

## Competing interests

The authors declare no competing interests.
