## [Peer Review File · Nature Communications]

Reviewers' Comments:

Reviewer #1:

Remarks to the Author:

This manuscript describes a strategy and software for the simultaneous measurement of (super-resolution) images and the determination of the mobility and brightness in each pixel using correlation spectroscopy.

Each of the methods used here was previously reported, and therefore the novelty in this work is in the combination of these approaches using a single, well-performing and presumably suitably user-friendly software tool developed by the authors.

Overall there is a clear appeal in such a tool. Image correlation has not been used up to its full potential, nor has it been applied to all of the many areas where it could be useful, partly due to the lack of suitable tools. This software is capable of infusing additional information in measurements at comparatively modest expense. In that sense it is a solid investigation that merits support.

However, while I certainly see the value of this approach, I do have concerns regarding the overall reliability of this tool, which would require more investigation. Overall the work would require major revisions before publication.

The key weakness of this work is its reliance on SRRF, an imaging technique that is not based on sound theory and that lacks reliability. This can be seen in figure 2b, which is not plausibly a version of figure 2a at higher resolution. I am including an image file that identifies a few particular regions where the SRRF image is not consistent with the TIRF image. A more detailed observation would reveal more. The second way in which this can be seen is in the overall rounded, 'blob-like' appearance of the images, which do not line up with other observations of the actin cytoskeleton. Figure 3f also shows pervasive blobbing. This is intrinsic to the method and reflects its shaky foundations. For example, the same non-plausible 'blob-like' features can be seen in Figure 4 in the original SRRF paper, <https://www.nature.com/articles/ncomms12471.pdf>. Finally, the third way in which this can be seen is in figure S4, where the apparent SRRF resolution varies with the exposure time. This is clearly strange. One would expect that the noise may change as a result of changing the exposure time, but that is not what is seen here. Fundamentally, these demonstrate issues with the SRRF analysis itself. It does not provide a reliable method for super-resolution imaging, but rather fits in the pantheon of methods that may be used to obtain limited insights under limited conditions.

I understand that the authors have included SRRF due to the availability of a GPU-based implementation. However, the manuscript should be revised to acknowledge the reliability issues of SRRF, and should critically reflect on how this performance impacts the methodology. It would also require dropping claims of particular spatial resolutions. After all, a claim of a particular spatial resolution is meaningless if the imaging performance itself is in doubt. Much of the improvements claimed to be due to SRRF, such as thresholding of the image, could possibly have been developed by thresholding the TIRF image directly, which would have been more straightforward, easier to understand, and more reliable.

More specific comments:

In the very first paragraph the authors write that the "Spatial resolution depends on the number of photons collected and thus sets a lower limit on acquisition time. Molecular dynamics requires acquisition times shorter than the dynamics of interest and thus sets an upper limit. Since these two limits, in general, do not lead to an overlap region, the combination of spatiotemporal super-resolution microscopy has remained a challenge. Attempts in the past either restrict time resolution^{2,3} or concentration^{4,5}, require specialized instrumentation^{6,7}, or need specialized sample labeling^{8,9}."

In fact, this entire line of reasoning is flawed since the authors address these different limitations by binning the images to generate synthetic longer-exposure images. This is not an original solution developed in this paper, and therefore the authors should tone down this argument. Also see the similar remark in the next paragraph, "such attempts in an imaging mode have been hampered by lack of strategies that can bridge the limitations imposed by spatial and temporal resolution requirements". Clearly a well-working strategy has been around for a long time!

On line 167 the authors discuss limitations in the N&B analysis, due to the intensity filtering. This reasoning is unclear and should be expanded, perhaps as an SI note.

What is the added value of Figure 2h? If the authors want to emphasize the fibers more clearly then they could use a form of color mapping that encodes both brightness and D, such as the HSV-based display that is used in the visualization of FRET ratio images.

Why are the fits in the bottom panels of 2g and 2h more visually spread out for h, yet the COV is lower?

The entire argument linking COV to SRRF is cast into doubt due to the limitations of SRRF. To what extent can the same goal be achieved using intensity-based thresholding (possibly local segmentation) on the TIRF image?

Reviewer #2:

Remarks to the Author:

The manuscript "SRRF 'n' TIRF - FCS: Simultaneous spatiotemporal super-resolution microscopy" by Sankaran et al. aims to combine the spatial resolution obtained by super-resolution microscopy with the temporal resolution of single molecule fluorescence spectroscopy in a single data set using GPU based post processing.

I can fully support the authors' claim that achieving both sufficient spatial and temporal resolution currently requires 'mutually exclusive experimental strategies'. Additionally, the authors provide a ready-to-use plugin that can be implemented on a wide range of microscopy setups. Hence, the work does not only address the current need of a better spatiotemporal understanding of biological systems, but also provides an innovative and easily applicable solution. Therefore, I find the presented approach novel and relevant for publication in Nature Communications.

The key novelty of this new approach is to 1) obtain a super-resolved SRRF layer from an imaging FCS data set by temporal binning and 2) correlating this additional information with the dynamic information obtained by imaging FCS or N&B analysis. Unfortunately, the work cannot fully convince with respect to the first point, since the presented SRRF data seems to be still prone to artifacts even at optimized conditions (for a more specific elaboration on this see Question 1). Since the SRRF information is used as a spatial mask for the dynamic analysis, any spatiotemporal correlation obtained by this procedure seems to be questionable. This argument is further supported by the observation that the usage of the SRRF masks does not substantially improve the quality of the dynamic analysis when compared to the TIRF signal and even yields somewhat contradictory results in some cases (see Question 2). This may also be the reason for the results of the two color combined FCS SRRF experiments not being very conclusive, and therefore further control experiments would be required to state the biological implications this prominently (see Question 3).

Due to the given arguments, I consider the manuscript in the present state not ready for publication in Nature Communications. I suggest the authors to refocus their work by improving on the following aspects:

Remarks & Questions

The data shown in Figure S4 does not support that for the authors' final choice of SRRF parameters (SRRF pixel size 24 nm, integration time 200 ms) the super-resolution rendered image is free of artifacts. However, due to the contrast settings, this might be difficult to judge. Hence, I would like to ask the authors to provide the images (Figure S4 a-e) as individual TIF files. As a guide to the eye, an overlay of Figure S4a, b and d in different colors would be helpful. Additionally, some of the actin fibers in the SRRF image in Figure 3f are not visible in the TIRF image in Figure 3e (i.e. the curly structures instead of the elongated ones). In fact, they look very similar to the artifacts shown in Figure S10. The authors should comment on this.

The on-fibre cases 1-3 in Figure 2f yield almost the same diffusion coefficient D of $\sim 0.8 \text{ } [\mu\text{m}]^2/\text{s}$ but the COV increases with distance from the fibre as judged by the SRRF signal. Especially for case 3 in Figure 2f one would already expect a high ratio of (fast) freely diffusing LifeAct within the volume used for calculation and thus an increase in D . Why can this not be seen in the average diffusion coefficient $D = 0.75 \text{ } [\mu\text{m}]^2/\text{s}$ which actually constitutes the lowest value of cases 1-3? More specific, one would expect a mix of a few slow particles with a high SNR (on-fibre) and many fast particles with a low SNR (off-fibre) contributing to the signal in case 3. How does this affect the obtained diffusion coefficient for this volume if a one species model is used? It seems that the diffusion coefficient tends to be dominated by the species with high SNR? Following this argument, to me it seems that the value $D = 0.75 \text{ } [\mu\text{m}]^2/\text{s}$ of case 3 (including a higher proportion of freely diffusing LifeAct) contradicts the value $D = 1.38 \text{ } [\mu\text{m}]^2/\text{s}$ of case 5?

The authors claim that their work indicates that 'partitioning of EGFR is primarily determined by EGFR-membrane interactions, possibly sub-resolution clustering and inter-EGFR interactions but independent of EGFR-actin interactions'. On the other hand, they state that they cannot 'exclude the possibility that our spatial resolution for the dynamics, even with SRRF filtering, is not sufficient to detect more subtle EGFR/actin interactions'. In my opinion, further control experiments would be required by, e.g. modifying the membrane dynamics or its coupling to the actin cytoskeleton, to make such a prominent claim (as in the abstract). Another option would be the demonstration of the technique on a less complex system where a correlation between the diffusion properties or oligomeric states to the presence of another molecule is already well established.

For N&B and SRRF a posteriori temporal binning was used for both sCMOS and EMCCD cameras. Does it improve SNR (and hence the localization precision) in both cases? If yes, is it due to the noise being dominated by the read-out noise (and not by the shot noise) for the used acquisition cycle time of 2 ms for both EMCCD and sCMOS? Could we also expect an increase in SNR by temporal binning in a shot noise limited regime? Why does the noise not simply add up?

The section 'Optimization for FCS and FCS diffusion law analysis' is not consistent with the content, which really describes the calibration of the PSF in order to be able to obtain results for diffusion dynamics from downstream FCS analysis (the authors could refer to the corresponding equation and specifically name the parameter). Furthermore, an unexperienced reader has to guess what happens in the left panels in Figure S2 (trying out varying PSF sizes until D is invariant with respect to binning?). The authors should elaborate on this in the caption.

In the same section, the reader is surprised by SLB experiments with diffusing 'fluorescent molecules', which are not further specified.

In line 180-181 the authors claim to show that in SRRF spatial resolution increases with time binning and decreasing pixel size, but they only show the time binning effect for a fixed SRRF pixel size of 24 nm. The same time binning series (2 ms, 20 ms, 200 ms) could be performed with another pixel size to illustrate this effect. (Question to Figure S4e: Was the original image first binned 2x2, and then analyzed with SRRF pixel size 48? Or the same field of view imaged without

2x magnification in front of the camera?)

The wording in the figure descriptions and the text is often misleading. For instance, In Figure 2g, 'Before SRRF mask; on-fibre SRRF and TIRF...'. To my understanding, under ideal, artifact-free conditions, all pixels in the diffusion map corresponding to the SRRF mask are a subset of all the pixels in the TIRF mask. A simpler wording such as 'TIRF mask' and the higher-resolved 'SRRF mask' would make this easier to follow.

It would be helpful to see the corresponding autocorrelation curves with respect to the given results for the 6 cases in Figure 2f (at least in the SI). For instance, does case 3 also show the ill-defined slow component?

Minor errors and unclear points:

Line 175: Sec 2.6 does not exist, do you mean 1.6?

Line 183: 'Further binning in time does not improve the FWHM but is prone to the creation of artefacts (Sec. 4 in the supplement)'

Is it not Sec. 3, i.e. Figs. S9-S11?

Table 1: SEM is defined as $SEM = STD/\sqrt{n}$ with n being the number of observations (i.e. n=6 in this case).

-Recalculation of the SEM of m_e with the values given in Table 1 yields $SEM(m_e) = 0.078$?

-Recalculation of the SEM of B^* with the values given in Table 1 yields $SEM(B^*) = 0.157$?

-Was eq. 20 in section 1.6.2.1 used instead, please clarify?

Line 578-580: 'The variance is typically affected by shot noise, and hence we used the covariance. This is referred to as G1 analysis. $Covariance(F(t), F(t + \Delta t)) = \langle F(t)F(t + \Delta t) \rangle$ where $F(t)$ is the fluorescence signal at time t and $F(t+\Delta t)$ is the fluorescence signal at time $t+\Delta t$ '

-So the covariance was used instead of the variance σ^2 ?

-If yes, at which time lag? I think this description is quite confusing for a reader who is not necessarily familiar with the 'G1 analysis'?

Reviewer #3:

Remarks to the Author:

Sankaran et al use a clever approach to combine super-resolution fluorescence microscopy (SRFM) with measurement approaches for fast temporal diffusion dynamics. Specifically they combine the recent SRRF SRFM approach with correlation analysis approaches on a TIRF microscope, and they employed multi-parameter analysis approaches to extract as much information from the same data set as possible. They exemplified their approach on investigating the super-resolved structure and dynamics of LifeAct-actin and epidermal growth factor receptor (EGFR). Especially, the two-color data on LifeAct and EGFR is impressive. In principle the authors take together well-known and -established approaches, but the novelty is the way of how this combination is achieved, which definitely deserves publication in Nature Communications.

- In the introduction, the authors could mention further tries towards super-resolved temporal dynamic measurements such as iMSD (Di Rienzo et al (2013) PNAS 110, 12307-12312), line-scanning STED-FCS (Honigmann et al (2014) Nature Communications 5, 5412), single-molecule displacement mapping (Xiang et al (2020) Nature Methods 17, 524-530), and the original SPT-SRFM work (Manley et al (2008) Nat Methods 5, 155-157).

- Mention a few more details (one sentence) on the way of the point spread determination (page 4, line 140).

- Any error bars in figures S1, S3, S6?

- General: Check consistent use of past and present.

- What do the authors mean by slow ill-defined component (page 5, line 208)? It is explained later

on, but the use of "ill-defined" at this instance is confusing.

- The parameter number of particles per pixels shows up the first time on page 8 almost out of nowhere. Would be good to introduce and define it better. Also, give some reasons why this value improves with SNR and SRRF.

- What is the r-parameter on page 8 (somehow got lost here)? Would be good to introduce it properly (yet, maybe I missed it).

- Page 9, line 353: What lipid domains? This is the first time this shows up and therefore it is confusing.

- Page 9, lines 364ff: This paragraph is slightly confusing – trapping is indeed clearly responsible for lower EGFR mobility as reasoned by the positive diffusion law intercept. The hypothesis of "This implies either that EGFR oligomers are 367 located in a more viscous membrane environment, e.g. lipid domains³⁴ or clathrin-coated pits, 368 or that we detect multiple EGFR molecules in larger complexes" should be brought afterwards and clearly as a hypothesis.

REVIEWER COMMENTS

Reviewer #1 (Remarks to the Author):

1. This manuscript describes a strategy and software for the simultaneous measurement of (super-resolution) images and the determination of the mobility and brightness in each pixel using correlation spectroscopy. Each of the methods used here was previously reported, and therefore the novelty in this work is in the combination of these approaches using a single, well-performing and presumably suitably user-friendly software tool developed by the authors. Overall there is a clear appeal in such a tool. Image correlation has not been used up to its full potential, nor has it been applied to all of the many areas where it could be useful, partly due to the lack of suitable tools. This software is capable of infusing additional information in measurements at comparatively modest expense. In that sense it is a solid investigation that merits support.
- We would like to thank Reviewer #1 for the positive evaluation and also the following critical comments, which we think has helped us in strengthening the manuscript.
2. However, while I certainly see the value of this approach, I do have concerns regarding the overall reliability of this tool, which would require more investigation. Overall the work would require major revisions before publication. The key weakness of this work is its reliance on SRRF, an imaging technique that is not based on sound theory and that lacks reliability.
- We agree with the reviewer that SRRF is prone to artefacts as we show in the supplement (Fig. S8), a discussion which we extended now in response to these comments. Actually our approach does not rely on SRRF and can be combined with any other computational super-resolution or deconvolution approach. SRRF is not a requirement. But the importance of this approach is the combination of temporal and spatial information that mutually confirm and corroborate each other. We have now included a section showing that FCS can be used to validate SRRF data and remove artefacts (Figs S6-S8, Supplementary note 4). The mutual consistency of the computational super-resolution method and the dynamics method provides a better estimate of dynamics parameters and allows corrections of possible structural artefacts. The advantage of computational super-resolution methods, including SOFI, SRRF, 3B etc. is that they do not need any special acquisition modalities but can be applied to the same data as the spectroscopic techniques measuring dynamics. Other super-resolution techniques can of course also be combined with spectroscopy techniques (e.g. STED and FCS, SMLM and single particle tracking etc.) but they typically need specialized instrumentation or sample conditions that are more constrained and less accessible to general users.

The first paragraph in the discussion has been rewritten as

“It is important to note that, although here we have used SRRF, simultaneous multiparametric spatiotemporal microscopy is also possible with other computational super resolution tools, e.g. 3B analysis¹, SOFI² or deconvolution methods³ since they do not need any special acquisition modalities but can be applied to the same data as the spectroscopic techniques measuring dynamics. The mutual consistency of the computational super-resolution method and the dynamics method provides a better estimate of dynamics parameters and allows corrections of possible structural artefacts.”

3. This can be seen in figure 2b, which is not plausibly a version of figure 2a at higher resolution. I am including an image file that identifies a few particular regions where the SRRF image is not consistent with the TIRF image. A more detailed observation would reveal more. The second way in which this can be seen is in the overall rounded, 'blob-like' appearance of the images, which do not line up with other observations of the actin cytoskeleton. Figure 3f also shows pervasive blobbing. This is intrinsic to the method and reflects its shaky foundations. For example, the same non-plausible 'blob-like' features can be seen in Figure 4 in the original SRRF paper, <https://www.nature.com/articles/ncomms12471.pdf>.

- We would like to thank the reviewer for these comments as it allowed us to fine tune the data treatment further. We agree with the reviewer that there are differences between the TIRF image and the SRRF image. As in other computational but also some experimental super-resolution methods, there is a trade-off between resolution or localization precision and feature retention and artefact creation. Sources of artefacts include double localization⁴ in single molecule localization microscopy (SMLM) and those that are caused due to illumination⁵, preparation⁶ and photobleaching⁷. In computational techniques this can be due to incomplete knowledge of the point spread function (PSF) or incorrect parameter selection⁸. To reduce these problems we have now used the TIRF images and the FCS data to correct SRRF artefacts. In the following we discuss the issues of loss of TIRF features and appearance of SRRF artefacts.

Loss of TIRF features

We have numbered 7 different regions provided by the reviewer (Fig RR1). All regions except region 6 are of low intensity in TIRF and the fibres appear either broken or are missing in SRRF (Fig. S4a and d). This is a result of a trade-off between FWHM reachable in SRRF and feature retention. Parameters which retain most of the TIRF features in SRRF lead to an increase in FWHM of bright fibres in SRRF. This is a general problem for computational super-resolution or deconvolution algorithms. The advantage in our application now is that the users can choose parameters that lead to images that can be corroborated by the dynamics measurements of FCS.

We use gradient smoothing in our SRRF analysis which results in a smoothing effect on the image and has been shown to improve the precision of radially⁹. Usually, the gradients are calculated using a 1×3 and 3×1 differentiation kernel pair in x and y directions respectively. Instead, the use of gradient smoothing leads to an increase in the size of the kernel pair used for calculating the gradient to 3×5 and 5×3. The FWHM of the intensity profile across the fibre was found to be 59 ± 5 nm (Fig. S4d). The improvement in precision of radially was at the cost of decreased sensitivity to high spatial frequencies⁹. We observe that without gradient smoothing, if the gradients are calculated using 1×3 and 3×1 differentiation kernel pair, the FWHM of the actin fibre increases to 72 ± 6 nm (Fig. RR1). But there is a larger retention of features from TIRF as marked by regions 1-5 and 7 in the figure. This shows that the choice of parameters used in SRRF analysis determines the proportion of feature retention from TIRF and the FWHM of the fibre in SRRF. So gradient smoothing provides a better starting point for further refinement by FCS filtering as described in the manuscript (Page 22, methods section, data analyses → SRRF).

Fig. RR1: Comparison of TIRF and SRRF images to delineate “loss of TIRF features” and “SRRF artefacts”: The yellow arrows 1-7 in all the images mark the positions indicated by the reviewer. **(a)** TIRF image of a CHO-K1 cell labeled with Lifeact-EGFP at 200 \times magnification. In panels (b) and (c), images from SRRF with and without gradient smoothing are shown on the left. Normalized intensity profiles across actin fibres at three different positions (indicated on the cell image by magenta, grey, and orange lines) are shown on the right. All the reported values in the figure are the FWHM of Gaussian fits to the intensity profiles. **(b)** SRRF image and histogram generated with gradient smoothing after time binning of the 2 ms data to 200 ms. The FWHM (mean \pm SD) of Gaussian Fits to the normalized intensity profiles are 59 ± 5 nm for the fibre. **(c)** SRRF image and histogram generated without gradient smoothing after time binning of the 2 ms data to 200 ms. The FWHM (mean \pm SD) of Gaussian Fits to the normalized intensity profiles are 72 ± 6 nm for the fibre. The pixel size after magnification is 120 nm (refer to Fig. S4 and Table S2). The scale bars shown in yellow measure 2.5 μ m in images (a) and (b).

SRRF artefacts and FCS based correction strategy

Region 6 in Fig. RR1, shows blob-like artefacts in the SRRF fibres which are a result of SRRF and have no equivalence in the TIRF image. We refer to this as artefactual regions. The choice of SRRF parameters determines the amount of artefacts. It is a trade-off between improvement in FWHM of intensity across actin fibre and the creation of artefacts. In this paper, we simultaneously perform FCS and SRRF and hence we utilize the diffusion coefficients derived from FCS to correct for such SRRF artefacts (Supplementary note-4). Hence we have chosen SRRF parameters which lead to an improvement in FWHM of intensity across actin fibre. We then categorized the various SRRF artefacts and the steps where they were removed. SRRF artefacts are of two types, off-fibre artefacts and clusters (Fig. S8).

Firstly, we use the TIRF image as a mask to remove pixels in between fibres. We then use the existing SRRF image as a mask to remove off-fibre pixels from the diffusion map. Since we are performing a combined SRRF-TIRF-FCS, we have now utilized the diffusion coefficient obtained from FCS to remove artefacts from SRRF. We call the TIRF-SRRF- D thresholded diffusion map the D consistency (D_C) map since this map leads to a consistent interpretation of dynamics and structural measurements. Using the D_C map as a mask, we filter the SRRF images to remove artefacts. The strategy is explained in the schematic (Fig. S6). Fig. 2g-i shows the results utilizing the strategy.

We can see the artefactual fibre region marked 6 has been removed in the SRRF consistency image (SRRF_C) in Fig. 2i. Region-6 being an off-fibre artefact is removed by TIRF mask. The red box in Fig. 2i shows that the corrected SRRF image has reduced artefacts when compared to the original map shown in Fig. 2c (shown below). This can also be observed in Fig. S13 as shown in response to question 4.

Based on our strategy, we find that off-fibre artefacts are removed by the application of the TIRF mask and clusters are removed by D thresholding. Apart from removing SRRF artefacts, this filtering strategy also led to an improvement in the coefficient of variation of the estimate of diffusion coefficient when compared to the unfiltered image.

4. Finally, the third way in which this can be seen is in figure S4, where the apparent SRRF resolution varies with the exposure time. This is clearly strange. One would expect that the noise may change as a result of changing the exposure time, but that is not what is seen here. Fundamentally, these demonstrate issues with the SRRF analysis itself.

- The situation for SRRF is similar to single molecule localization microscopy (SMLM). In SMLM identifiable areas, where a fluorophore is found, are fit with a 2D Gaussian to locate the particle. The uncertainty of the localization depending on pixel size, background and signal, is then indicated by a smooth 2D Gaussian (or another approximation of the PSF)¹⁰. Thus the SNR is encoded in the width of that distribution but the distribution itself is smooth and does not show noise. This is the same for SRRF which uses gradients (and potentially gradient smoothing see section 2.3 in the supplement) to locate structures. The final plot is a probability distribution (covariance). So we agree that the noise on the plot of the fibre cross sections is not representative of the noise of the image. That is rather encoded in the precision of the localization and the width of the plot. Theoretically, averaging over n frames leads to \sqrt{n} improvement in SNR. Denoting the n pixel counts in an image as a random variable x ,

Without post-processing	Averaging over n frames
$Mean(x_i)=\mu$	$Mean(x_i)=\mu$
$Var(x_i)=\sigma^2$	$Var\left(\frac{\sum_{i=1}^n x_i}{n}\right)=\frac{1}{n^2}\sum_{i=1}^n Var(x_i)=\frac{n\sigma^2}{n^2}=\frac{\sigma^2}{n}$
$SD=\sigma$	$SD=\frac{\sigma}{\sqrt{n}}$
$SNR = \frac{\mu}{\sigma}$	$SNR = \frac{\mu}{\frac{\sigma}{\sqrt{n}}} = \frac{\sqrt{n}\mu}{\sigma}$

In the case of Fig. S4, the SNR of 2 ms is 1.5. Upon time-averaging to 20 ms and 200 ms, the SNR increases to 3.1 and 3.7 respectively. The improvement in SNR with increase in averaging time leads to an improved in localization of the structures and thus the FWHM.

Time resolution	Intensity of Lifeact image	Intensity of background image	SNR
Raw data-2 ms	151.3 ± 39.3	92.7 ± 9.5	1.5
Time averaged data-20 ms	151.3 ± 19.2	92.7 ± 3.1	3.1
Time averaged data-200 ms	151.4 ± 15.8	92.8 ± 1.1	3.7

To better demonstrate that the final SRRF consistency image is a possible location of the actual fibres, we overlap and compare with the original TIRF image (Fig. S13). The merged image shows that both artefacts and fibres are included in the original SRRF image. However, after filtering, the artefacts are removed and only the fibres are retained.

Fig. S13: Consistency pixels are localized on the fibres in the TIRF image: (a) TIRF image of a CHO-K1 cell labeled with Lifact-EGFP at 200X magnification. This image was thresholded (>140 counts) and converted into a mask. (b) The uncorrected SRRF image at 200 ms binning is shown at the top. This image was thresholded (>0) and merged with the TIRF mask, and is shown at the bottom. The TIRF, SRRF and correlated pixels are colored green, magenta and white, respectively. The off-fibre artefacts in magenta are visible clearly. (c) The SRRF_c image at 200 ms binning is shown at the top. The merge of this image with the TIRF mask is shown at the bottom. The TIRF, SRRF and correlated pixels are colored green, magenta and white, respectively. Since the artefacts have been removed in the SRRF_c image, only the SRRF fibres are visible and are clearly on the TIRF fibres. Note that there is some loss of fibrillar features also during the creation of the SRRF_c image. The scale bar shown in yellow in all the images measures 2.5 μm .

This should not distract from the fact that SRRF has its pitfalls as we try to make now also clearer in the manuscript. But as we note, it is the consistency of the SRRF mask with the FCS data that increases our confidence in the results. And this would be true for other computational super-resolution or deconvolution methods.

5. It does not provide a reliable method for super-resolution imaging, but rather fits in the pantheon of methods that may be used to obtain limited insights under limited conditions. I understand that the authors have included SRRF due to the availability of a GPU-based implementation. However, the manuscript should be revised to acknowledge the reliability issues of SRRF, and should critically reflect on how this performance impacts the methodology.

➤ SRRF allows the better localization of fluorescent features and relies on radial gradients and correlations to reduce noise. Localization super-resolution approaches fit 2D Gaussians or similar functions to single localizations. By using radial gradients SRRF relaxes the constraint to the 2D fits but at the same time gives up exact localization as can be reached by SMLM. This comes at the advantage that SRRF is less restricted by concentrations and does not need to switch fluorophores on and off. So we agree that it is not on a par with

other super-resolution techniques. But it allows better localization under certain circumstances especially under conditions which are compatible with imaging FCS, improving both spatial and dynamics estimates.

We discuss now in the supplement (supplementary note 4) more of these problems and point out artefacts of SRRF (off-fibre artefacts and aggregates diffusing through the sample as identified by FCS), and how we can correct them using imaging FCS.

6. It would also require dropping claims of particular spatial resolutions. After all, a claim of a particular spatial resolution is meaningless if the imaging performance itself is in doubt.

- We agree that resolution vs localization in microscopy is an important issue and is often confounded. We try now to make a clear distinction between resolution and localization. Previously we had utilized FWHM as a metric to quantify the SRRF performance. Hence we had stated 60 nm as the smallest size of spatial features attainable and thus the best localization of a fibre. Now we utilize peak to peak distance (p2p) and Fourier ring correlation (FRC) to quantify the resolution as ~100 nm (Fig. 2e), which we think is in line with other publications on the subject¹¹⁻¹³. The abstract currently reads

“To achieve both, we implement a GPU-supported, camera-based measurement strategy that highly resolves spatial structures (~100 nm), temporal dynamics (~2 ms), and molecular brightness from the exact same data set.”

One of the sentences in the manuscript (page 6) has also been corrected to the form below:

“The use of SRRF led to a more accurate estimation of the localization of actin fibres.”

7. Much of the improvements claimed to be due to SRRF, such as thresholding of the image, could possibly have been developed by thresholding the TIRF image directly, which would have been more straightforward, easier to understand, and more reliable.

- We agree with the reviewer that even simple intensity thresholding by TIRF improves D estimation as it localizes fibres, albeit with less precision than what we can reach with SRRF. TIRF masking itself does not provide the localization of structures and thus cannot filter the D map as efficiently as SRRF as seen from our results. Briefly, both TIRF and SRRF images were thresholded with the aim to retain fibres to create masks. The precision of the D estimate of thresholded SRRF ($0.67 \pm 0.45 \mu\text{m}^2/\text{s}$, COV-66%) as a mask is better than that of TIRF only ($0.58 \pm 0.47 \mu\text{m}^2/\text{s}$, COV-81%) (Fig. S12). The large error with the D_{TIRF} is due to the presence of fibre border pixels which are not removed by the use of the TIRF mask. The SRRF mask leads to more efficient removal of border pixels leading to a better estimation of D . However, TIRF can be used as a mask to remove SRRF artefacts. Therefore, we use now both the TIRF and SRRF masks to remove off-fibre artefacts and localize fibres respectively. To reduce user induced bias at the two steps of filtering, we utilize user defined intensity thresholding for TIRF and subsequently utilize any pixel with SRRF amplitude > 0 as a valid pixel. The combination of TIRF and SRRF is more efficient at removing pixels with low artefactual diffusion coefficients when compared to either TIRF or SRRF masking (Fig. S7d). We thank the reviewer for insisting on this point as it allowed us to improve the filtering strategy.

8. More specific comments: In the very first paragraph the authors write that the "Spatial resolution depends on the number of photons collected and thus sets a lower limit on acquisition time. Molecular dynamics requires acquisition times shorter than the dynamics of interest and thus sets an upper limit. Since these two limits, in general, do not lead to an overlap region, the combination of spatiotemporal super-resolution microscopy has remained a challenge. Attempts in the past either restrict time resolution^{2,3} or concentration^{4,5}, require specialized instrumentation^{6,7}, or need specialized sample labeling^{8,9}." In fact, this entire line of reasoning is flawed since the authors address these different limitations by binning the images to generate synthetic longer-exposure images. This is not an original solution developed in this paper, and therefore the authors should tone down this argument. Also see the similar remark in the next paragraph, "such attempts in an imaging mode have been hampered by lack of strategies that can bridge the limitations imposed by spatial and temporal resolution requirements". Clearly a well-working strategy has been around for a long time!
- The paragraph is meant as a motivation for the work. To bring spatial and dynamics measurements together one needs to find some compromise which in this case is simply fast acquisition with small pixel sizes. This will come with reduced SNR and increased data that need to be treated, both issues we try to address in this manuscript. But we do not make any priority claim. So we did not change this paragraph. We hope the reviewer agrees to this perspective since we think it is important to explain how this manuscript was conceived.

The phrase in the next paragraph reads now in full:

“While simultaneous multi-parametric fluorescence detection (MFD) has been established for point-measurements^{14,15}, simultaneous parameter estimations in an imaging mode have been limited¹⁶ and have been hampered by lack of strategies that can bridge the limitations imposed by spatial and temporal resolution requirements and by the computationally expensive data evaluation procedures required to treat the large data sets.”

The double issue of recording data with high temporal and spatial resolution in an imaging mode and treat that data in an acceptable time, ideal real time, with standard instrumentation is an important issue, which motivated this work. But of course we agree that there are different approaches to the problem and we try to cite them.

Researchers up to now have pursued to optimize either spatial or temporal resolution, that typically did not allow simultaneous high spatial and temporal resolution. This was mainly a technical issue. We circumvent this problem by taking the smallest pixels available and the best time resolution and then bin either in time or space to optimize the signal for spatial or temporal resolution. We agree as well that this is not necessarily an original approach. But we think it explains to the reader why this approach is used. It is really fast cameras and GPUs that allow us to do both now at the same time with acceptable evaluation times of the large data sets involved. We are still limited by photons per particle since for better resolution we need more photons per particle in a small pixel ideally, and we need larger pixels to obtain sufficient number of photons for fast measurements.

9. On line 167 the authors discuss limitations in the N&B analysis, due to the intensity filtering. This reasoning is unclear and should be expanded, perhaps as an SI note.

- Fig. S3 has been modified explaining the intensity filtering as well in Supplementary note 2. Fluorescence images obtained after various thresholding are shown in the figure.

The following text is added to the supplement:

“A threshold is chosen to separate the cellular and non-cellular regions in the image. The effect of varying the threshold is shown in Fig. S3d-h. Insufficient thresholding retains some of the cellular exterior while excessive thresholding leads to reduction of cellular areas.”

Fig. S3: Number and Brightness analysis – Effects of exposure time, measurement time and intensity thresholding on brightness. (a) and (b) The mean brightness of PMT-mApple and PMT-mApple₂ is plotted against the total measurement time, for different exposure times (2 ms, 10 ms, 20 ms). (c) The brightness ratio of PMT-mApple₂/PMT-mApple is plotted against total measurement time, for different exposure times (2 ms, 10 ms, 20 ms). Each point is an average of 3 different cell measurements for both PMT-mApple and PMT-mApple₂. The number of pixels in each cell at each acquisition time is different due to differential intensity filtering in each case, but each cell had at least 1850 and at most 6250 valid pixels. (d) TIRF image of a CHO-K1 cell labeled with EGFR-mApple after 20 ms time binning at 100× magnification. In panels (e)-(h), example B maps after different intensity

thresholding are shown on the left. The corresponding histogram of the B values are shown on the right. **(e)** B_E map and histogram without thresholding. All pixels, including the background, are retained on the B_E map. The histogram shows two peaks with the background pixels as a prominent peak on the left. The average B_E value is underestimated. **(f)** B_E map and histogram after insufficient thresholding of ≥ 3000 counts. Some pixels from the background are still retained on the B_E map and appear as a small peak on the left in the histogram. The average B_E value is underestimated. **(g)** B_E map and histogram after appropriate thresholding of ≥ 7000 counts. Almost all pixels from the background are eliminated on the B_E map and a single peak is seen in the histogram. The average B_E value is representative of the cell alone. **(h)** B_E map and histogram after excessive thresholding of ≥ 13000 counts. Many pixels from the cell are eliminated on the B_E map. The average B_E value is overestimated. Mean \pm SEM is reported here. The scale bars shown in yellow measure $5 \mu\text{m}$ in images (d)-(h).

10. What is the added value of Figure 2h? If the authors want to emphasize the fibres more clearly then they could use a form of color mapping that encodes both brightness and D , such as the HSV-based display that is used in the visualization of FRET ratio images.

➤ Fig. 2i in the current version (Fig. 2h in the previous version) is now the final consistency diffusion map which is obtained after applying the TIRF, SRRF masks and thresholding the diffusion coefficient. This figure referred to as the D consistency map enables the computation of the diffusion coefficient along the fibres only, removing the contributions of the off-fibre area and aggregates. The D_C map and SRRF $_C$ image are validated to each other by retaining only the common pixels between them.

11. Why are the fits in the bottom panels of 2g and 2h more visually spread out for h, yet the COV is lower?

➤ Figs. 2g and 2h in the previous version display the evolution of ACFs after D thresholding and D-SRRF thresholding respectively. In the current submission with the modified strategy, the evolution of ACFs is shown in Figs. 2g-i. In the case of Figs. 2g-i, we see that there is a difference in both amplitude and width. N and D variation in ACFs were convoluted, so we normalized the ACFs now to show the variability in width only (Fig. S11). S11c shows the set of ACFs after the application of TIRF mask while S11e shows the set of ACFs corresponding to the D_C map. We can see that the curves are more visually spread out in Fig. S11c when compared to Fig. S11e. The COV of the estimate of diffusion coefficients obtained from visually spread out curves in (c) is 81% while the COV of less spread out curves shown in (e) is 49%.

Fig. S11: Normalized ACFs for the different thresholded D maps. (a) The normalized ACFs for the D map without thresholding ($D_L = 0.48 \pm 0.54 \mu\text{m}^2/\text{s}$) are shown. (b) The normalized ACFs for the SRRF-masked D map ($D_{\text{SRRF}} = 0.60 \pm 0.47 \mu\text{m}^2/\text{s}$) are shown. (c) The normalized ACFs for the TIRF-masked D map ($D_{\text{TIRF}} = 0.58 \pm 0.47 \mu\text{m}^2/\text{s}$) are shown. (d) The normalized ACFs for the D map after applying both TIRF mask and SRRF mask ($D_{\text{TIRF-SRRF}} = 0.68 \pm 0.45 \mu\text{m}^2/\text{s}$) are shown. (e) The normalized ACFs for the D_C map generated after D thresholding ($D \geq 0.175 \mu\text{m}^2/\text{s}$; $D_C = 0.80 \pm 0.39 \mu\text{m}^2/\text{s}$) are shown. Mean \pm SD is reported here.

12. The entire argument linking COV to SRRF is cast into doubt due to the limitations of SRRF. To what extent can the same goal be achieved using intensity-based thresholding (possibly local segmentation) on the TIRF image?

- The precision of the D estimate of using the SRRF mask ($0.67 \pm 0.45 \mu\text{m}^2/\text{s}$, COV-66%) is better than that of TIRF mask only ($0.58 \pm 0.47 \mu\text{m}^2/\text{s}$, COV-81%) (Fig. S12). The large error of the D estimate due to TIRF is due to the presence of fibre border pixels which are not removed by the use of the TIRF mask.

TIRF alone does not provide any super-resolved images. For this a computational approach is required. We chose here SRRF but as pointed out it could be other computational methods. And we think this is the strength of the manuscript, namely that it provides dynamics and high-resolution images with a computational time on the order of the measurements time or even faster, and that with standard equipment.

Please refer answer to Q7 for a detailed discussion on the importance of SRRF in the filtering strategy.

Reviewer #2 (Remarks to the Author):

13. The manuscript "SRRF 'n' TIRF - FCS: Simultaneous spatiotemporal super-resolution microscopy" by Sankaran et al. aims to combine the spatial resolution obtained by super-resolution microscopy with the temporal resolution of single molecule fluorescence spectroscopy in a single data set using GPU based post processing. I can fully support the authors' claim that achieving both sufficient spatial and temporal resolution currently requires 'mutually exclusive experimental strategies'. Additionally, the authors provide a ready-to-use plugin that can be implemented on a wide range of microscopy setups. Hence, the work does not only address the current need of a better spatiotemporal understanding of biological systems, but also provides an innovative and easily applicable solution. Therefore, I find the presented approach novel and relevant for publication in Nature Communications. The key novelty of this new approach is to 1) obtain a super-resolved SRRF layer from an imaging FCS data set by temporal binning and 2) correlating this additional information with the dynamic information obtained by imaging FCS or N&B analysis.

- We would like to thank Reviewer #2 for the positive comments.

14. Unfortunately, the work cannot fully convince with respect to the first point, since the presented SRRF data seems to be still prone to artifacts even at optimized conditions (for a more specific elaboration on this see Question 1).

- We have now performed extensive characterization of artefacts in SRRF and identified factors that contribute to the artefacts. We initially use the TIRF mask to remove pixels between fibres from the diffusion map. We therefore used the SRRF images, which have smaller pixel size and thus better fibre localization compared to TIRF images, as a mask to better identify those pixels that correspond to fibre positions. Apart from that, since we performed a combined SRRF-FCS measurement, we have utilized diffusion coefficients obtained from FCS to threshold diffusion maps. This TIRF-SRRF- D thresholded diffusion map we call the D consistency (D_C) map. We now use the D_C map to mask the SRRF image and we see that this approach leads to the elimination of artefacts (Figs. 2i, S6-S8). It is

now discussed in supplementary note 4 and also discussed in the answer to the reviewer #1 (Q3) in page 2.

15. Since the SRRF information is used as a spatial mask for the dynamic analysis, any spatiotemporal correlation obtained by this procedure seems to be questionable. This argument is further supported by the observation that the usage of the SRRF masks does not substantially improve the quality of the dynamic analysis when compared to the the signal and even yields somewhat contradictory results in some cases (see Question 2). This may also be the reason for the results of the two color combined FCS SRRF experiments not being very conclusive, and therefore further control experiments would be required to state the biological implications this prominently (see Question 3). Due to the given arguments, I consider the manuscript in the present state not ready for publication in Nature Communications. I suggest the authors to refocus their work by improving on the following aspects:

Remarks & Questions

The data shown in Figure S4 does not support that for the authors' final choice of SRRF parameters (SRRF pixel size 24 nm, integration time 200 ms) the super-resolution rendered image is free of artifacts.

- We acknowledge the presence of artefacts at SRRF pixel size 24 nm, integration time 200 ms in Fig. S4. Using TIRF masking, SRRF masking and utilizing diffusion coefficient from FCS as parameters for thresholding, we identify pixels which are localized on the fibres. The diffusion map obtained after TIRF, SRRF, and D thresholding is referred to as D consistency (D_C) map. Using the D_C map as a mask on the SRRF image, we demonstrate that artefactual regions in SRRF are removed (Fig 2i, red box).
16. However, due to the contrast settings, this might be difficult to judge. Hence, I would like to ask the authors to provide the images (Figure S4 a-e) as individual TIF files.
- We have attached the individual TIF files along with this submission.
17. As a guide to the eye, an overlay of Figure S4a, b and d in different colors would be helpful.
- A new panel-f consisting of the overlay of Fig. S4a, b and d has been added. The white pixels in this figure are common to all the three images. The yellow pixels are those common only to TIRF and SRRF at 2 ms. The TIRF pixels not present in any of the SRRF images are shown in green. The red and magenta pixels are artefacts created in the SRRF image which are effectively removed by our filtering strategy. In addition, also refer to question 4 and Fig. S13 where the TIRF and SRRF masks are overlaid.

f

TIRF and SRRF
maps merge
18. Additionally, some of the actin fibres in the SRRF image in Figure 3f are not visible in the TIRF image in Figure 3e (i.e. the curly structures instead of the elongated ones). In fact, they look very similar to the artifacts shown in Figure S10. The authors should comment on this.

➤ Yes, such curly structures are referred to as artefacts in this manuscript. We have employed a filtering strategy to remove such artefacts. Fig. 4f (earlier Fig. 3f) and Fig. S16 (earlier Fig. S10) are images after filtering removing areas containing such curly structures. A detailed description of the filtering strategy is provided in the supplementary note 4. Briefly, we create a mask using the original TIRF image and then use it to filter the SRRF image in case of Figs. 4f and Figs. S16.

19. The on-fibre cases 1-3 in Figure 2f yield almost the same diffusion coefficient D of ~ 0.8 $[\mu\text{m}]^2/\text{s}$ but the COV increases with distance from the fibre as judged by the SRRF signal. Especially for case 3 in Figure 2f one would already expect a high ratio of (fast) freely diffusing Lifeact within the volume used for calculation and thus an increase in D . Why can this not be seen in the average diffusion coefficient $D = 0.75$ $[\mu\text{m}]^2/\text{s}$ which actually constitutes the lowest value of cases 1-3?

➤ The effect of freely diffusing Lifeact on D at 2×2 bin is not seen due to the poor SNR of the off-fibre ACFs. As shown in Fig. S9b, the autocorrelation curves obtained from off-fibre exhibit a poor signal to noise ratio at 2×2 bin and upon fitting yield an artefactual low diffusion coefficient. The fast diffusion of freely diffusing particles is captured only upon binning 4×4 as seen in Fig. S9d.

We would also like to highlight that we have modified our strategy to filter the diffusion maps and relabeled the areas. Currently the diffusion coefficient on the fibre (D_C) is estimated to be $0.80 \pm 0.39 \mu\text{m}^2/\text{s}$ (Fig. 3-box 3). The diffusion coefficient of the area with the largest contribution from off-fibre (D_O) is $0.41 \pm 0.46 \mu\text{m}^2/\text{s}$ (Fig. 3-box 1).

20. More specific, one would expect a mix of a few slow particles with a high SNR (on-fibre) and many fast particles with a low SNR (off-fibre) contributing to the signal in case 3. How does this affect the obtained diffusion coefficient for this volume if a one species model is used?

➤ Yes, Fig. 3-box 1 currently (case 3 earlier) is a mixture of on fibre and off-fibre particles. Given the differences in SNR between the on-fibre and the off-fibre, the two kinds of particles cannot be reliably resolved by a two particle fit. Among the three cases, box 1 in the current version has the highest COV of 112%. The COV of the other two cases are 81%

and 49%. The increased COV of case 1 is due to the presence of two different populations of diffusing particles.

21. It seems that the diffusion coefficient tends to be dominated by the species with high SNR? Following this argument, to me it seems that the value $D = 0.75 \text{ } [\mu\text{m}]^2/\text{s}$ of case 3 (including a higher proportion of freely diffusing Lifeact) contradicts the value $D = 1.38 \text{ } [\mu\text{m}]^2/\text{s}$ of case 5?

- Yes, the diffusion coefficient tends to be dominated by the species with high SNR. At 2×2 bin, the ACFs of freely diffusing particles have a lower SNR leading to an artefactual diffusion coefficient (D_B) of $0.41 \pm 0.46 \text{ } \mu\text{m}^2/\text{s}$. On the contrary, increasing the bin size to 4×4 leads to an improvement in SNR leading to an increase in diffusion coefficient (D_B) to $1.73 \pm 1.46 \text{ } \mu\text{m}^2/\text{s}$. The difference in diffusion coefficient in the previous submitted version and the current version is due to the fact that we have modified our filtering strategy. We now employ a 3 step filtering strategy: TIRF mask, SRRF mask, and D thresholding. The diffusion coefficient of box 1 in the current version (case 3 in the earlier version) is found to be $0.41 \pm 0.46 \text{ } \mu\text{m}^2/\text{s}$.

It is to be noted that both binning and fitting are changed in this comparative study. A change of bin size from 2×2 to 4×4 and fitting only the fast component leads to a change of D_B from $0.41 \pm 0.46 \text{ } \mu\text{m}^2/\text{s}$ to $1.73 \pm 1.46 \text{ } \mu\text{m}^2/\text{s}$. Fitting the 4×4 fast component only for SRRF fibre (D_C) yields a diffusion coefficient of $1.26 \pm 0.44 \text{ } \mu\text{m}^2/\text{s}$ (box-7). The diffusion coefficient off-fibre is larger than the D estimated on fibre and also has an increased error compared to the estimate on fibre due to reduced SNR (Fig. S7c).

22. The authors claim that their work indicates that ‘partitioning of EGFR is primarily determined by EGFR-membrane interactions, possibly sub-resolution clustering and inter-EGFR interactions but independent of EGFR-actin interactions’. On the other hand, they state that they cannot ‘exclude the possibility that our spatial resolution for the dynamics, even with SRRF filtering, is not sufficient to detect more subtle EGFR/actin interactions’. In my opinion, further control experiments would be required by, e.g. modifying the membrane dynamics or its coupling to the actin cytoskeleton, to make such a prominent claim (as in the abstract). Another option would be the demonstration of the technique on a less complex system where a correlation between the diffusion properties or oligomeric states to the presence of another molecule is already well established.

- We have now added experiments with the addition of external agents which modify membrane dynamics by cholesterol removal (M β CD and COase) and depolymerize actin (LAT-A). We have also added experiments where we added the cognate ligand-EGF which is known to lead to oligomerization of EGFR.

EGF addition, known to form oligomers leads to an expected increase in brightness and a decrease in diffusion coefficient. It also leads to an increase in diffusion law intercept indicating that dynamic formation of EGFR clusters leads to an increase in diffusion law intercept.

We find that cholesterol removal leads to decrease in diffusion coefficient, increase in brightness and diffusion law intercept. The increased diffusion law intercept and brightness and reduced diffusion coefficient is attributed to the dynamic formation of EGFR clusters. Cholesterol removal leads to receptors being spatially less constrained and hence leads to

an increase in probability of cluster formation¹⁷. Apart from EGFR clustering, partitioning into cholesterol independent domains also leads to an increase in intercept when compared to resting state¹⁸.

Depolymerization of actin leads to conspicuous absence of long, thick actin filaments in TIRF and SRRF images. A decrease in diffusion coefficient and an increase in FCS diffusion law intercept is observed. The intercept in diffusion law is affected both by domain trapping and influence of cytoskeleton. Trapping leads to an increase in the intercept while influences from the cytoskeleton leads to a decrease in the intercept. The $B_{E, LAT-A}$ remained similar to resting state, indicating that the EGFR oligomerization state is not disturbed. Hence the observed increase in the diffusion law intercept is only due to the fact that the effects of the cytoskeleton are relieved upon treatment of LAT-A.

Overall, this multi-parametric investigation reveals that EGFR exists as monomers and dimers in cholesterol dependent and independent domains which are not influenced by actin cytoskeleton directly. Upon removal of cholesterol, EGFR is freed from cholesterol dependent domains leading to an increase in propensity to oligomerize in cholesterol independent domains.

Figures S19-S22 have been added to the supplement. Table 2 has been added to the manuscript. New text explaining the experimental data has been added to the manuscript.

23. For N&B and SRRF a posteriori temporal binning was used for both sCMOS and EMCCD cameras. Does it improve SNR (and hence the localization precision) in both cases? If yes, is it due to the noise being dominated by the read-out noise (and not by the shot noise) for the used acquisition cycle time of 2 ms for both EMCCD and sCMOS? Could we also expect an increase in SNR by temporal binning in a shot noise limited regime? Why does the noise not simply add up?

- N&B and SRRF are both techniques with fundamentals in correlation spectroscopy and thus remove noise that is random and thus uncorrelated (white noise). Averaging over n frames is typically performed while analysing fluorescence correlation spectroscopy data using a multi tau correlator scheme in FCS¹⁹. Averaging or binning over n frames leads to improvement in \sqrt{n} times in the Signal to Noise ratio. The standard deviation of the signal obtained after averaging in bins is always less than the standard deviation without binning. Hence binning leads to improved estimate of the ratio of the dimer to monomer fraction since each contributing factor in the ratio is estimated more precisely.

The signal to noise ratio was quantified using the expression below for both cameras²⁰.

$$\frac{S}{N} = \frac{QE * P}{ENF * \sqrt{QE * P + ReadNoise^2 + Exposuretime * I_{dark}}}$$

P is the photon flux per pixel during the measurement time. ENF is the excess noise factor. In the case of EMCCD, it is the multiplicative noise quantified as square root of two. In the case of sCMOS, the value of ENF is 1.

Fig. RR2: Performance of EMCCD and sCMOS cameras: **(a)** Variation of signal to noise ratio with photons per pixel. **(b)** Same as in (a) plotted only upto 100 photon per pixel and also plotting an EMCCD camera with no readout and dark noise. **(c)** A plot of SNR relative to that of an ideal camera. The red and green markers in (a)-(c) represent the photons per pixel for our measurements using EMCCD and sCMOS, respectively.

	iXon^{EM}+ 860	Sona 4.2B-11
QE	0.925	0.917
ENF	1.41	1
Read Noise	0.2 per pixel	1.6 per pixel
I_{dark}	0.002/s per pixel	0.4/s per pixel
Exposure time	2 ms	2 ms

The custom camera in blue in (b) in this figure is an imaginary EMCCD camera with $QE = 1$, Read Noise = 0, Dark noise = 0, $ENF = 1.414$. In the case of even an ideal EMCCD camera, the presence of multiplicative noise makes it impossible to become an ideal sensor. We observe that in the case of the EMCCD camera used in this paper, when SNR is plotted against photon flux, the line is parallel to that of the line of an ideal sensor indicating that the EMCCD camera used is shot noise limited and not read out noise limited. As seen in the table, the readout noise is very low and is equal to 0.2 electrons per pixel.

Unlike the EMCCD, the profile of the sCMOS camera, clearly shows, two regimes, a shot noise limited regime and a readout noise limited regime. In Fig. RR2-C, the % difference in SNR between SONA and the ideal sensor is plotted for various levels of photons per pixel. For less than 161 photons per pixel, there is at least 5% difference in SNR between ideal sensor and SONA. Hence this region is referred to as the readout noise limited regime. For photons per pixel > 161 , it is referred to as shot noise limited regime.

In the case of EMCCD, 10 MHz, 14-bit EM, the sensitivity of the EMCCD is 15.8 electrons per A/D count. At a gain of 300, after subtracting the background, for the data shown using measurements with Lifeact in dual channel using an EMCCD, there are 316 counts on average within the measurement time of 2 ms which corresponds to a photon flux of 18 photons/pixel. This is in the multiplicative noise limited regime as explained earlier.

In the case of dual channel measurements with Lifeact using a sCMOS the photon flux was found to be 33 photons per pixel in 2 ms. The CMOS sensitivity is 0.61 electrons per A/D count. This is in the readout noise limited regime. In both, the readout noise case of the

sCMOS and multiplicative noise limited regime of the EMCCD, time-averaging leads to an improvement in the determination of FWHM in N&B.

A similar effect is seen in the case of EGFR in the case of N&B. In the case of EMCCD, binning to 20 ms from 2 ms leads to an improvement of SNR from 6.3 to 7.6 whereas in the case of sCMOS it was from 1.7 to 3.9. As seen in this table, the SNR is improved for both cameras.

Time binning	SRRF		N&B	
	EMCCD	sCMOS	EMCCD	sCMOS
2 ms	2.8	3.1	6.3	1.7
20 ms	3.9	3.9	7.6	3.9
200 ms	4.0	4.0		

24. The section ‘Optimization for FCS and FCS diffusion law analysis’ is not consistent with the content, which really describes the calibration of the PSF in order to be able to obtain results for diffusion dynamics from downstream FCS analysis (the authors could refer to the corresponding equation and specifically name the parameter). Furthermore, an unexperienced reader has to guess what happens in the left panels in Figure S2 (trying out varying PSF sizes until D is invariant with respect to binning?). The authors should elaborate on this in the caption.

- The section has been renamed to “Calibration of Imaging FCS and FCS diffusion law analysis”. We have added the equation in the supplementary note-1 and added a description about the procedure to estimate the PSF. The Fig. S2 is now part of the supplementary note-1. A reference is made to Sec. 1.4 in the supplement which describes the left panels in Fig. S2.

The following text has been added to the manuscript describing the procedure to estimate PSF.

*“The measurement of molecular dynamics requires a time resolution about ten times faster than the characteristic process under investigation²¹, in our case the time a particle needs to diffuse across the observation area of one pixel. This observation area is determined by the convolution of the pixel area and the point spread function (PSF) of the microscope²¹. The pixel size of a camera is known but the PSF must be experimentally determined²². This can either be done by scanning a probe small compared to the PSF over a pixel to determine the observation area²³ or it can be measured using a freely diffusing sample, e.g. a lipid probe in a one component supported lipid bilayer, as used here²². For a freely diffusing particle the diffusion coefficient (D) is independent of the observation area and D should be constant at different spatial pixel binning. This, however, is only the case if the correct PSF is used in the calculation of the pixel observation area and thus D . Therefore, we vary the size of a Gaussian PSF until it leads to a constant D . In our case, the $1/e^2$ radius of the Gaussian PSF was found to be 272 nm and 364 nm for the measurements using 488 nm and 561 nm lasers, respectively (Fig. S2). Using these lasers, the measured diffusion coefficients of lipids diffusing in a fluorescently labeled supported lipid bilayer consisting of 1,2-dioleoyl-*sn*-glycero-3-phosphocholine (DOPC) are found to be 1.97 ± 0.91 and $2.00 \pm 0.34 \mu\text{m}^2/\text{s}$ (Supplementary note 1, Table S2), similar in range to those reported in the literature²⁴. For a single pixel the observation area can now be calculated to be $0.48 \mu\text{m}^2$ at 565 nm. With a D of $\sim 2 \mu\text{m}^2/\text{s}$, this corresponds to an average transition time²¹ of a*

particle through the pixel observation area of 60 ms, more than ten times slower than our recording speed.”

The following sentence is also added to the caption:

“The procedure to obtain the PSF calibration plot is described in Sec. 1.4 in the supplement. Briefly, the diffusion coefficients at various bin sizes are determined for different values of the PSF. The PSF which yields a diffusion coefficient independent of the bin area is the PSF of the system.”

25. In the same section, the reader is surprised by SLB experiments with diffusing ‘fluorescent molecules’, which are not further specified.

- We are sorry for this oversight and have rewritten the section and explain this now properly. Diffusion of labelled lipid in supported lipid bilayers were being measured. The phrase ‘fluorescent molecules’ has been replaced with ‘lipids’ and the sentence has been rewritten as:

“Using these lasers, the measured diffusion coefficients of lipids diffusing in a fluorescently labeled supported lipid bilayer consisting of 1,2-dioleoyl-sn-glycero-3-phosphocholine (DOPC) are found to be 1.97 ± 0.91 and 2.00 ± 0.34 $\mu\text{m}^2/\text{s}$ (Supplementary note 1, Table S2), similar in range to those reported in the literature²⁴.”

26. In line 180-181 the authors claim to show that in SRRF spatial resolution increases with time binning and decreasing pixel size, but they only show the time binning effect for a fixed SRRF pixel size of 24 nm. The same time binning series (2 ms, 20 ms, 200 ms) could be performed with another pixel size to illustrate this effect. (Question to Figure S4e: Was the original image first binned 2x2, and then analyzed with SRRF pixel size 48? Or the same field of view imaged without 2x magnification in front of the camera?)

- It is the same field of view imaged without 2× magnification in front of the camera. We also show it for another magnification now with a SRRF pixel size of 48 nm. The FWHM is reduced from 256 ± 15 nm to 124 ± 35 nm. The increase in SRRF pixel size from 24 nm to 48 nm leads to the increase in FWHM from 59 ± 5 (Fig. S4) to 124 ± 35 nm (Fig. S5).

The following sentences have been added to the manuscript:

“Upon time binning to 200 ms, this improves by 3.2-fold to 59 ± 5 nm. In the case of a pixel size of 240 nm, time binning to 200 ms led to a two-fold improvement in the FWHM from 256 ± 15 nm to 127 ± 22 nm (Fig. S5).”

Fig. S5: Time binning improves spatial resolution in SRRF at pixel size of 240 nm: In panels (a)-(d), images from TIRF or SRRF as indicated are shown on the left. Normalized intensity profiles across actin fibres at three different positions (indicated by magenta, grey, and orange lines on the cell image) are shown on the right. All the reported values in the figure are the FWHM of Gaussian fits to the intensity profiles. **(a)** TIRF image of a CHO-K1 cell labeled with Lifeact-EGFP at 100 \times magnification (same cell shown in Fig. 4). **(b)** SRRF image of the 2 ms data shown in image (a). **(c)** and **(d)** show the SRRF images after time binning of the 2 ms data shown in image (a) to 20 ms and 200 ms, respectively. (The FWHM (mean \pm SD) of Gaussian Fits to the normalized intensity profiles from each image is as follows: (a) 524 ± 64 nm, (b) 256 ± 15 nm, (c) 159 ± 2 nm, and (d) 127 ± 22 nm. The pixel sizes reported are after magnification (refer Table 3). The scale bars shown in yellow measure 5 μ m in images (a)-(d).

27. The wording in the figure descriptions and the text is often misleading. For instance, In Figure 2g, ‘Before SRRF mask; on-fibre SRRF and TIRF...’. To my understanding, under ideal, artifact-free conditions, all pixels in the diffusion map corresponding to the SRRF mask are a subset of all the pixels in the TIRF mask. A simpler wording such as ‘TIRF mask’ and the higher-resolved ‘SRRF mask’ would make this easier to follow.

Yes, SRRF pixels are a subset of TIRF pixels and we also agree that the original naming was somewhat complex. Hence we simplified the nomenclature now by using “on fibre” for SRRF identified pixels, “border pixels” for pixel that seem to be fibres on TIRF but not

on SRRF, and “off-fibre” pixels that are neither identified by SRRF nor TIRF to be on fibres.

28. It would be helpful to see the corresponding autocorrelation curves with respect to the given results for the 6 cases in Figure 2f (at least in the SI). For instance, does case 3 also show the ill-defined slow component?

➤ We have prepared an extra figure and added it to the supplement (Fig. S10).

No, box 1 in the current version (case 3 in the earlier version) does not show the ill-defined slow component at 2×2 binning explicitly since the pixels in box 1 contain pixels which are localized on TIRF pixels as well.

Fig. S10: Representative ACFs from fibre and off-fibre areas. This figure shows the representative ACFs for the different colored boxes shown in the schematic of Fig. 3. The data from boxes 1, 3 and 4 are from 2x2 binning and fitted for the full measurement time of 100 seconds. The data from boxes 5, 7 and 8 are from 4x4 binning and only the fast diffusion component is fitted. (a) Representative ACF from box 1 (border pixels; $D_B = 0.91 \mu\text{m}^2/\text{s}$). (b) Representative ACF from box 3 (on-fibre; $D_C = 0.83 \mu\text{m}^2/\text{s}$). (c) Representative ACF from box 4 (off-fibre; low SNR). (d) Representative ACF from box 5 (border pixels; $D_B = 1.93 \mu\text{m}^2/\text{s}$). (e) Representative ACF from box 7 (on-fibre; $D_C = 1.38 \mu\text{m}^2/\text{s}$). (f) Representative ACF from box 8 (off-fibre; $D_O = 2.83 \mu\text{m}^2/\text{s}$). (g) Normalized representation of all the ACFs shown in (a)-(f).

29. Minor errors and unclear points:

Line 175: Sec 2.6 does not exist, do you mean 1.6?

- Yes, it is Sec 1.6. This has been corrected.

30. Line 183: ‘Further binning in time does not improve the FWHM but is prone to the creation of artefacts (Sec. 4 in the supplement)’ Is it not Sec. 3, i.e. Figs. S9-S11?

- Yes, it is Sec. 3. This has been corrected.

31. Table 1: SEM is defined as $SEM = \text{STD}/\sqrt{n}$ with n being the number of observations (i.e. $n=6$ in this case).

- Recalculation of the SEM of m_e with the values given in Table 1 yields $SEM(m_e) = 0.078$?
- Recalculation of the SEM of B^* with the values given in Table 1 yields $SEM(B^*) = 0.157$?
- Was eq. 20 in section 1.6.2.1 used instead, please clarify?

- Yes, Eq. 20 (Eq. 19 in the current version) in section 1.6.2.1 was used to calculate the SEM. The following text is added below Table 1:

“The SEM was calculated using Eq. 19 in the supplement.”

32. Line 578-580: ‘The variance is typically affected by shot noise, and hence we used the covariance. This is referred to as G1 analysis. $\text{Covariance}(F(t), F(t + \Delta t)) = \langle F(t)F(t + \Delta t) \rangle$ where $F(t)$ is the fluorescence signal at time t and $F(t+\Delta t)$ is the fluorescence signal at time $t+\Delta t$ ’

- So the covariance was used instead of the variance σ^2 ?
- If yes, at which time lag? I think this description is quite confusing for a reader who is not necessarily familiar with the ‘G1 analysis’?

- Yes, the covariance was used instead of the variance.

The time lag was the best possible time resolution. In our case, it was 2 ms, which is smaller than the average time a particle needs to diffuse across the pixel. A D of $\sim 0.2 \mu\text{m}^2/\text{s}$ corresponds to an average transition time of a particle through the pixel observation area of 600 ms.

Reviewer #3 (Remarks to the Author):

33. Sankaran et al use a clever approach to combine super-resolution fluorescence microscopy (SRFM) with measurement approaches for fast temporal diffusion dynamics. Specifically they combine the recent SRRF SRFM approach with correlation analysis approaches on a TIRF microscope, and they employed multi-parameter analysis approaches to extract as much information from the same data set as possible. They exemplified their approach on investigating the super-resolved structure and dynamics of Lifeact-actin and epidermal growth factor receptor (EGFR). Especially, the two-color data on Lifeact and EGFR is impressive. In principle the authors take together well-known and -established approaches, but the novelty is the way of how this combination is achieved, which definitely deserves publication in Nature Communications.

➤ We would like to thank Reviewer #3 for the helpful and constructive comments.

34. In the introduction, the authors could mention further tries towards super-resolved temporal dynamic measurements such as iMSD (Di Rienzo et al (2013) PNAS 110, 12307-12312), line-scanning STED-FCS (Honigmann et al (2014) Nature Communications 5, 5412), single-molecule displacement mapping (Xiang et al (2020) Nature Methods 17, 524–530), and the original SPT-SRFM work (Manley et al (2008) Nat Methods 5, 155-157).

➤ The references have been added and are numbered 15, 9, 10 and 11 respectively.

35. Mention a few more details (one sentence) on the way of the point spread determination (page 4, line 140).

➤ The following text has been added to the manuscript:

“The measurement of molecular dynamics requires a time resolution about ten times faster than the characteristic process under investigation²¹, in our case the time a particle needs to diffuse across the observation area of one pixel. This observation area is determined by the convolution of the pixel area and the point spread function (PSF) of the microscope²¹. The pixel size of a camera is known but the PSF must be experimentally determined²². This can either be done by scanning a probe small compared to the PSF over a pixel to determine the observation area²³ or it can be measured using a freely diffusing sample, e.g. a lipid probe in a one component supported lipid bilayer, as used here²². For a freely diffusing particle the diffusion coefficient (D) is independent of the observation area and D should be constant at different spatial pixel binning. This, however, is only the case if the correct PSF is used in the calculation of the pixel observation area and thus D. Therefore, we vary the size of a Gaussian PSF until it leads to a constant D. In our case, the $1/e^2$ radius of the Gaussian PSF was found to be 272 nm and 364 nm for the measurements using 488 nm and 561 nm lasers, respectively (Fig. S2). Using these lasers, the measured diffusion coefficients of lipids diffusing in a fluorescently labeled supported lipid bilayer consisting of 1,2-dioleoyl-sn-glycero-3-phosphocholine (DOPC) are found to be 1.97 ± 0.91 and $2.00 \pm 0.34 \mu\text{m}^2/\text{s}$ (Supplementary note 1, Table S2), similar in range to those reported in the literature²⁴. For a single pixel the observation area can now be calculated to be $0.48 \mu\text{m}^2$ at 565 nm. With a D of $\sim 2 \mu\text{m}^2/\text{s}$, this corresponds to an average transition time of a particle through the pixel observation area of 60 ms, more than ten times slower than our recording speed.”

36. Any error bars in figures S1, S3, S6?

➤ The error bars are included in Fig. S1 and S3. Fig. S7c (previously Fig. S6) is the number of particles at a particular pixel. Hence it does not have error bars.

37. General: Check consistent use of past and present.

➤ Corrected. We describe the experiments in past tense and conclusions in present tense.

38. What do the authors mean by slow ill-defined component (page 5, line 208)? It is explained later on, but the use of “ill-defined” at this instance is confusing.

➤ The phrase has been removed at this instance.

39. The parameter number of particles per pixels shows up the first time on page 8 almost out of nowhere. Would be good to introduce and define it better. Also, give some reasons why this value improves with SNR and SRRF.

- A reduction in SNR leads to an increase in the estimate of the number of particles per pixel²⁵ (equivalent to a lower FCS amplitude). We observed that pixels with large N (low SNR) can be attributed to pixels not directly on the fibres, while small N (large SNR) can be attributed to pixel on top of actin fibres.

The following text is added to the supplement.

“We observed that off-SRRF fibre pixels have a reduced ACF amplitude when compared to pixels located on the SRRF fibre which indicates that the off-fibre ACFs have a reduced SNR when compared to those located on fibres²⁵. We quantified the amplitude by calculating the number of particles per pixel (N). The number of particles which is inversely proportional to the amplitude is lowest on the fibre and increases in the neighboring pixels (Fig. S7c).”

40. What is the r -parameter on page 8 (somehow got lost here)? Would be good to introduce it properly (yet, maybe I missed it).

- The following sentence is now added:

“In the case of N&B, the ratio between brightness of EGFR to that of brightness of the monomer (r) was found to be 1.6. In this case, a simple dimer model is not reasonable anymore.”

41. Page 9, line 353: What lipid domains? This is the first time this shows up and therefore it is confusing.

- The sentence has been corrected.

“The positive intercepts for all entries in Table S6 indicate that EGFR undergoes hindered diffusion due to confinement in lipid domains (both cholesterol dependent and cholesterol independent) independent of the presence or absence of cytoskeleton or clusters (Fig. S15, Table S6).”

42. Page 9, lines 364ff: This paragraph is slightly confusing – trapping is indeed clearly responsible for lower EGFR mobility as reasoned by the positive diffusion law intercept. The hypothesis of “This implies either that EGFR oligomers are 367 located in a more viscous membrane environment, e.g. lipid domains³⁴ or clathrin-coated pits, 368 or that we detect multiple EGFR molecules in larger complexes” should be brought afterwards and clearly as a hypothesis.

- We have rearranged the paragraph. Also the sentence currently reads:

“Based on these observations, we hypothesize that either EGFR oligomers are located in a more viscous membrane environment, e.g. lipid domains²⁶ or clathrin-coated pits, or that we detect multiple EGFR molecules in larger complexes^{27,28}.”

References

- 1 Cox, S. *et al.* Bayesian localization microscopy reveals nanoscale podosome dynamics. *Nature Methods* **9**, 195-200, doi:10.1038/nmeth.1812 (2012).
- 2 Dertinger, T., Colyer, R., Iyer, G., Weiss, S. & Enderlein, J. Fast, background-free, 3D super-resolution optical fluctuation imaging (SOFI). *Proceedings of the National Academy of Sciences* **106**, 22287-22292, doi:10.1073/pnas.0907866106 (2009).
- 3 Sibarita, J.-B. in *Microscopy Techniques: -/-* (ed Jens Rietdorf) 201-243 (Springer Berlin Heidelberg, 2005).
- 4 Li, H. & Vaughan, J. C. Switchable Fluorophores for Single-Molecule Localization Microscopy. *Chemical Reviews* **118**, 9412-9454, doi:10.1021/acs.chemrev.7b00767 (2018).
- 5 GEORGIADES, P., ALLAN, V. J., DICKINSON, M. & WAIGH, T. A. Reduction of coherent artefacts in super-resolution fluorescence localisation microscopy. *Journal of Microscopy* **264**, 375-383, doi:10.1111/jmi.12453 (2016).
- 6 Whelan, D. R. & Bell, T. D. M. Image artifacts in Single Molecule Localization Microscopy: why optimization of sample preparation protocols matters. *Scientific Reports* **5**, 7924, doi:10.1038/srep07924 (2015).
- 7 Godin, A. G., Lounis, B. & Cognet, L. Super-resolution microscopy approaches for live cell imaging. *Biophys J* **107**, 1777-1784, doi:10.1016/j.bpj.2014.08.028 (2014).
- 8 Parthasarathy, R. Rapid, accurate particle tracking by calculation of radial symmetry centers. *Nature Methods* **9**, 724-726, doi:10.1038/nmeth.2071 (2012).
- 9 Gustafsson, N. *et al.* Fast live-cell conventional fluorophore nanoscopy with ImageJ through super-resolution radial fluctuations. *Nature Communications* **7**, 12471, doi:10.1038/ncomms12471 (2016).
- 10 Thompson, R. E., Larson, D. R. & Webb, W. W. Precise Nanometer Localization Analysis for Individual Fluorescent Probes. *Biophys J* **82**, 2775-2783, doi:[https://doi.org/10.1016/S0006-3495\(02\)75618-X](https://doi.org/10.1016/S0006-3495(02)75618-X) (2002).
- 11 Zeng, Z., Ma, J., Xi, P. & Xu, C. Joint tagging assisted fluctuation nanoscopy enables fast high-density super-resolution imaging. *Journal of Biophotonics* **11**, e201800020, doi:10.1002/jbio.201800020 (2018).
- 12 Solomon, O., Mutzafi, M., Segev, M. & Eldar, Y. C. Sparsity-based super-resolution microscopy from correlation information. *Opt. Express* **26**, 18238-18269, doi:10.1364/OE.26.018238 (2018).
- 13 Han, Y. *et al.* Ultra-fast, universal super-resolution radial fluctuations (SRRF) algorithm for live-cell super-resolution microscopy. *Opt. Express* **27**, 38337-38348, doi:10.1364/OE.27.038337 (2019).
- 14 Weidtkamp-Peters, S. *et al.* Multiparameter fluorescence image spectroscopy to study molecular interactions. *Photochemical & Photobiological Sciences* **8**, 470-480, doi:10.1039/B903245M (2009).
- 15 Esposito, A. & Venkitaraman, A. R. Enhancing Biochemical Resolution by Hyperdimensional Imaging Microscopy. *Biophys J* **116**, 1815-1822, doi:<https://doi.org/10.1016/j.bpj.2019.04.015> (2019).
- 16 Di Rienzo, C., Gratton, E., Beltram, F. & Cardarelli, F. Fast spatiotemporal correlation spectroscopy to determine protein lateral diffusion laws in live cell membranes.

- Proceedings of the National Academy of Sciences* **110**, 12307-12312, doi:10.1073/pnas.1222097110 (2013).
- 17 Pike, L. J. & Casey, L. Cholesterol levels modulate EGF receptor-mediated signaling by altering receptor function and trafficking. *Biochemistry* **41**, 10315-10322, doi:10.1021/bi025943i (2002).
- 18 Bag, N., Huang, S. & Wohland, T. Plasma Membrane Organization of Epidermal Growth Factor Receptor in Resting and Ligand-Bound States. *Biophys J* **109**, 1925-1936, doi:10.1016/j.bpj.2015.09.007 (2015).
- 19 Schaetzl, K. & Peters, R. *Noise on multiple-tau photon correlation data*. Vol. 1430 PWL (SPIE, 1991).
- 20 Mullan, A. *Calculating the Signal to Noise Ratio of a Camera*, <<https://andor.oxinst.com/learning/view/article/ccd-signal-to-noise-ratio>> (2019).
- 21 Sankaran, J., Bag, N., Kraut, R. S. & Wohland, T. Accuracy and Precision in Camera-Based Fluorescence Correlation Spectroscopy Measurements. *Analytical Chemistry* **85**, 3948-3954, doi:10.1021/ac303485t (2013).
- 22 Bag, N., Sankaran, J., Paul, A., Kraut, R. S. & Wohland, T. Calibration and Limits of Camera-Based Fluorescence Correlation Spectroscopy: A Supported Lipid Bilayer Study. *ChemPhysChem* **13**, 2784-2794, doi:10.1002/cphc.201200032 (2012).
- 23 Krieger, J. W. *et al.* Imaging fluorescence (cross-) correlation spectroscopy in live cells and organisms. *Nature Protocols* **10**, 1948-1974, doi:10.1038/nprot.2015.100 (2015).
- 24 Guo, L. *et al.* Molecular diffusion measurement in lipid Bilayers over wide concentration ranges: A comparative study. *ChemPhysChem* **9**, 721-728, doi:10.1002/cphc.200700611 (2008).
- 25 Sankaran, J. *et al.* Single microcolony diffusion analysis in *Pseudomonas aeruginosa* biofilms. *npj Biofilms and Microbiomes* **5**, 35, doi:10.1038/s41522-019-0107-4 (2019).
- 26 Sergé, A., Bertaux, N., Rigneault, H. & Marguet, D. Dynamic multiple-target tracing to probe spatiotemporal cartography of cell membranes. *Nature Methods* **5**, 687-694, doi:10.1038/nmeth.1233 (2008).
- 27 Saffarian, S., Li, Y., Elson, E. L. & Pike, L. J. Oligomerization of the EGF receptor investigated by live cell fluorescence intensity distribution analysis. *Biophys J* **93**, 1021-1031, doi:10.1529/biophysj.107.105494 (2007).
- 28 Nagy, P. *et al.* Activation-dependent clustering of the erbB2 receptor tyrosine kinase detected by scanning near-field optical microscopy. *Journal of Cell Science* **112**, 1733 (1999).

Reviewers' Comments:

Reviewer #1:

Remarks to the Author:

The manuscript by Sankaran et al. has seen considerable work in these revisions. My main concern with the original version was the reliability of the SRRF imaging, an aspect that was not only identified by Reviewer 2 but that is now also explicitly mentioned by the authors, who agree that the technique is prone to introduce artifacts.

The authors try to address this by mentioning that their approach could be combined with other computational frameworks, which is true, but such frameworks are not part of this contribution.

The proposed solution is to adapt the analysis to mitigate the effects of such artifacts. This gives rise to a rather baroque strategy in which the FCS, SRRF, and TIRF data are repeatedly used to mask out various parts of the other images (figure S6). Such an analysis is unconventional, has many moving parts, and goes through great efforts for no reason other than to suppress the artifacts in the SRRF imaging. I have strong reservations about any technique where something as basic as acquiring an image requires such extensive 'cleaning' operations. This is the primary data that provides the starting point for actual investigations, and reliability issues at that early stage will cast doubt onto the entire effort.

Note also that the SRRF procedure not only generates spurious signals, but also eliminates valid signals (see e.g. arrow 2 in Fig. RR1b). Such valid signals cannot be recovered by masking.

In Fig. RR1 the authors focus on a specific parameter which appears to introduce artifacts, the 'gradient smoothing'. Disabling gradient smoothing may appear to reduce some of the artifacts, though at the end of the page the authors remark that "So gradient smoothing provides a better starting point for further refinement by FCS filtering". Why?

Suppose that we give the authors the benefit of the doubt, and assume that the correction strategy described here works suitably well on this data. What if the end user acquires data that looks entirely different, and that does not have such a clear-cut distinction between fiber and non-fiber parts of the image, or where the signal levels are lower? Note that the effects are hard to predict since the SRRF artifacts depend on the various settings and the specifics of the data (see for instance how easy it is to generate spurious features in Fig. S4 and Fig. S17).

While I support the spirit of the work of the authors, I can hardly escape from having serious reservations regarding the reliability of this strategy, and I am not convinced that it has the robustness desired for a general-purpose tool. If I was considering this tool for my own purposes then I would most likely stick to the TIRF and FCS data, and forget about the SRRF data entirely. And would I in fact lose anything much, since the FCS data is diffraction-limited anyway? The SRRF data is largely biased towards the bright features of the sample, so if desired I think I could do something similar but more robust by putting some more thought into the thresholding of the TIRF image (in the process also keeping those regions of the image that the SRRF calculation has improperly excluded).

The authors do comment on the thresholding with the TIRF image, though I find their reasoning unclear. They argue that "TIRF masking itself does not provide the localization of structures" (page 7 in the reply to the reviewers). But the whole point of any imaging is to provide the localization of structures! The authors continue that "TIRF can be used as a mask to remove SRRF artefacts". But how could it be used to decide which SRRF features are artifacts if it is incapable of providing the localization of structures?

Sadly, upon taking all of this in consideration I regret that I cannot support publication of this revised work in Nature Communications.

Reviewer #2:

Remarks to the Author:

The authors have invested substantial efforts in improving the analysis workflow and in addressing my initial remarks and concerns. They present two convincing and self-sustained approaches to handle prominent artifacts introduced by SRRF analysis: i) subtracting an intensity-thresholded TIRF mask from the SRRF image and ii) removal of pixels from the SRRF mask which yielded outlying FCS observables, caused by bright and very slow diffusing aggregates. With the latter, the authors were able to show that the dynamic information obtained via FCS analysis can lead to more robust structural information obtained via SRRF, which also constitutes an improvement of SRRF on its own. This reviewer acknowledges the thorough discussions and elaborations added to the main text and the SI. In its revised form, an interested reader is able to grasp the methodology and the technical details as well as potential pitfalls regarding artifacts. Furthermore, the discussion of cellular data has largely benefited from additional experiments quantifying the effects of cholesterol removal and addition of EGF ligand to the system. I can now recommend the manuscript for publication in Nature Communications and look forward to seeing the method added to the toolbox of imaging-based biological research.

Minor remarks:

Main

Line 321: ... the a more precise localization...

SI

Line 619: dynamics

Suppl. Note 3:

Unclear what is meant here: "Usually, the gradients are calculated using a 1×3 and 3×1 differentiation kernel pair in x and y directions respectively. Instead, the use of gradient smoothing leads to an increase in the size of the kernel pair used for calculating the gradient to 3×5 and 5×3 ." Pretty unclear what this means.

Reviewer #3:

Remarks to the Author:

The authors have well commented to most of my concerns as well have tackled those of the other referees – especially the other referees' point of potential bias and reliability indeed needed more attention. The manuscript has now significantly improved. I stick to my opinion that this manuscript deserves publication in NatComm.

One remaining issue from my side following my previous comment on the use of lipid domains. In my opinion the term "lipid domain" should be defined better – lipid domain on its on is too unspecific.

REVIEWER COMMENTS

We would like to thank all three reviewers for their evaluations and critical comments that we think make the article stronger. We own thanks especially to Reviewer #2 for making the effort and commenting on Reviewer #1 and providing constructive ways forward. But we also acknowledge the critical comments by Reviewer #1 who pointed out several ways of improvement. We therefore replied to and acted on all comments for completeness but focused especially on the overall comments of Reviewer #2.

Reviewer #1 (Remarks to the Author):

The manuscript by Sankaran et al. has seen considerable work in these revisions. My main concern with the original version was the reliability of the SRRF imaging, an aspect that was not only identified by Reviewer 2 but that is now also explicitly mentioned by the authors, who agree that the technique is prone to introduce artifacts.

We have developed now a strategy based on our multi-parametric measurements to reduce the artefacts leading to an improved consistency between structural and dynamics data.

Fig. 3: Schematic outlining the strategy of using SRRF image and D map to correct each other: The untreated $D_{L(2)}$ map, TIRF image and SRRF image of a CHO-K1 cell expressing Lifeact-EGFP serve as the starting files. First we create an intensity filtered TIRF mask which is then applied to the $D_{L(2)}$ map and SRRF image to remove the contributions from off-fibre artefacts generating the $D_{L,on(2)}^{TIRF}$ and $SRRF_L^{TIRF}$ images respectively. The $D_{L,on(2)}^{TIRF}$ map is then filtered by thresholding D values to remove the slow clusters. The resulting $D_{L,on(2)}^{TIRF,D}$ map is used as a mask to remove artefacts from the $SRRF_L^{TIRF}$ image to generate the $SRRF_L^{TIRF,D}$ image. For the dual-color measurements, the Lifeact SRRF image is filtered only by using the TIRF image as a mask. The orange box indicates an area that contained artefacts in the original $SRRF_L$ image and are now removed in the $SRRF_L^{TIRF,D}$ image.

For this purpose, we first filtered Imaging FCS data by retaining only pixels that can be attributed to structures in the TIRF image. However, this does not remove artefacts due to slow moving bright aggregates, as identified by FCS, and which are not actual structural features. By using only those pixels that contain diffusion coefficients consistent with diffusion *on fibres* we correct the SRRF image and remove artefacts that are created by mobile bright complexes. This strategy leads to reduction in the artefacts in the SRRF image. The strategy is now shown in Fig. 3. For clarity of notation, we provide the diffusion coefficient (D) of Lifeact with correction (*TIRF* filtered, D filtered, and/or *SRRF* filtered), location (*on* or *off* fibres) and pixel binning (2 or 4) denoted as $D_{L,location(bin)}^{correction\ methods}$ in Fig. 3.

The authors try to address this by mentioning that their approach could be combined with other computational frameworks, which is true, but such frameworks are not part of this contribution.

We have now added two other computational microscopy frameworks in our submission. They are deconvolution and super-resolution optical fluctuation imaging (SOFI) and demonstrate that in general, the strategy can be applied to other super-resolution or computational microscopy approaches.

Fig. 2: Multi-parametric analysis from a single channel fluorescence data set: (a) TIRF image of CHO-K1 cell expressing Lifact-EGFP at 200 \times magnification (120 nm pixel size). (b) The D map without thresholding ($D_{L(2)} = 0.48 \pm 0.54 \mu\text{m}^2/\text{s}$) is shown on the left. The COV is 113%. Representative ACFs are shown on the right. (c-e) Deconvolution image, SOFI image of order 2, SOFI image of order 4, respectively. (f) SRRF_L image (200 ms binning; refer Fig. S4) of the cell in (a). (g) Merge of the SRRF_L image and $D_{L(2)}$ map. The SRRF_L , $D_{L(2)}$ and correlated pixels are colored cyan, magenta and white, respectively. White pixels identify where the SRRF_L image and the diffraction-limited $D_{L(2)}$ map coincide. Cyan pixels show fibres in SRRF_L but no correlated D is found (SRRF artefacts). Magenta pixels show a D consistent with diffusion on fibres but no structure in SRRF. Due to the effect of diffraction limit of Imaging FCS, not all pixels consistent with diffusion on fibres are observed in the SRRF_L image. (h) Thickness (FWHM of the Gaussian fit) of the actin fibre before SRRF (magenta line in (a)) = 392 nm. Thickness of the actin fibre after deconvolution (green line in (c)) = 254 nm. Thickness of the actin fibre after SOFI-order 2 (red line in (d)) = 319 nm. Thickness of the actin fibre after SOFI-order 4 (grey line in (e)) = 217 nm. Thickness of the actin fibre after SRRF (orange line in (f)) = 77 nm. Refer to Fig. S6 for more details. (i) Enlarged views of yellow boxes in images (a) and (f). The intensity profile shows the actin fibre branching point (indicated by the orange line on the enlarged SRRF image). The peak-to-peak resolution is 96 nm at this point. The pixel sizes reported are after magnification (refer Table 3). The scale bars shown in yellow measure 2.5 μm in images (a)-(g), and 250 nm in (i). All values are reported as Mean \pm SD.

To keep with the idea of the manuscript we stayed with freely accessible approaches to allow users to easily implement their chosen solution. The data from SRRF and SOFI are provided in Figs. 2c-e. Interestingly, all computational microscopy techniques (deconvolution, SOFI and SRRF) showed artefacts, many similar and attributable to bright mobile particles. Among the three, SRRF yielded the lowest FWHM of the Gaussian fit to the intensity across the fibre. Hence we used SRRF as a case study in the rest of the article.

The proposed solution is to adapt the analysis to mitigate the effects of such artifacts. This gives rise to a rather baroque strategy in which the FCS, SRRF, and TIRF data are repeatedly used to mask out various parts of the other images (figure S6). Such an analysis is unconventional, has many moving parts, and goes through great efforts for no reason other than to suppress the artifacts in the SRRF imaging. I have strong reservations about any

technique where something as basic as acquiring an image requires such extensive 'cleaning' operations. This is the primary data that provides the starting point for actual investigations, and reliability issues at that early stage will cast doubt onto the entire effort. Note also that the SRRF procedure not only generates spurious signals, but also eliminates valid signals (see e.g. arrow 2 in Fig. RR1b). Such valid signals cannot be recovered by masking.

>> We have reproduced Fig. RR1 from our previous submission for clarification. In our previous submission, our SRRF data analysis pipeline consisted of a smoothing step. We had performed gradient smoothing since it led to the lowest FWHM of the Gaussian fit to the intensity across the fibre. However, the improved FWHM comes at the cost of losing certain fibre signals (for instance: arrow 2 in Fig. RR1b). We agree, that no amount of filtering will recover these fibres again. Hence, in this submission, throughout the study, we have not used gradient smoothing anymore. As seen in Fig. RR1c (same as Figs. S4d, S6b in the current submission), removal of gradient smoothing led to an increase in the FWHM of the Gaussian fit to the intensity across the fibre (59 ± 5 nm with gradient smoothing, 72 ± 7 nm without gradient smoothing). A comparison of Figs. RR1b (with smoothing) and RR1c (without smoothing) clearly reveals that removal of gradient smoothing has better fibre retention than the image created using gradient smoothing. The visibility of fibre marked by Arrow 2 in Fig. RR1c also highlights the better fibre retention without smoothing. However, any data evaluation on a raw image will require user defined settings that will lead to certain feature retention and other features to be dismissed as noise. By providing structural and temporal data and showing that the two types of data are consistent, we think we provide more certainty on the retained features and more information on the biological processes as the data can be clearly distinguished by location.

Fig. RR1: Comparison of TIRF and SRRF images to delineate “loss of TIRF features” and “SRRF artefacts”: (a) TIRF image of a CHO-K1 cell labeled with Lifect-EGFP at 200 \times magnification. In panels (b) and (c), images from SRRF with and without gradient smoothing are shown on the left. Normalized intensity profiles across actin fibres at three different positions (indicated on the cell image by magenta, grey, and orange lines) are shown on the right. All the reported values in the figure are the FWHM of Gaussian fits to the intensity profiles. (b) and (c) SRRF image and histogram generated with and without gradient smoothing respectively after time binning of the 2 ms data to 200 ms. The FWHM (mean \pm SD) of Gaussian Fits to the normalized intensity profiles are 59 ± 5 nm and 72 ± 7 nm for the fibre with and without gradient smoothing respectively. The pixel size after magnification is 120 nm (refer to Fig. S4 and Table S2). The scale bars shown in yellow measure 2.5 μ m.

In Fig. RR1 the authors focus on a specific parameter which appears to introduce artifacts, the 'gradient smoothing'. Disabling gradient smoothing may appear to reduce some of the

artifacts, though at the end of the page the authors remark that "So gradient smoothing provides a better starting point for further refinement by FCS filtering". Why?

>> We had used gradient smoothing earlier since utilizing gradient smoothing led to an improvement in FWHM of the Gaussian fits to the intensity profile across a fibre.

As discussed in the response to the previous question, disabling gradient smoothing led to improvement in fibre retention and hence we have currently disabled gradient smoothing in this submission.

Suppose that we give the authors the benefit of the doubt, and assume that the correction strategy described here works suitably well on this data. What if the end user acquires data that looks entirely different, and that does not have such a clear-cut distinction between fiber and non-fiber parts of the image, or where the signal levels are lower?

>> We agree with the reviewer, if no structural information is available it cannot be used to compare with dynamics data. In that case, only dynamics will be left to be measured and we will obtain only D , $N&B$ and diffusion law measurements for the different wavelength channels or any other parameters depending on the evaluation scheme for the time trace data. The question thus is about the signal to noise ratio (SNR) that is required to be able to distinguish structures. As pointed out in the manuscript (Introduction) the precision of localization of structures depends on the number of photons collected and thus on the SNR. In order to empirically ascertain the role of SNR in determining the efficiency of fibre retention after using our correction strategy, we evaluated the signal to noise ratio of three examples of bright and dim fibres on the TIRF image. All the six fibres (Fig. RR2, 3 bright-B1, B2, B3 and 3 dim-D1, D2, D3) are visible on the SRRF image obtained after filtering. The region marked as "OFF" was used to evaluate the intensity of the off-fibre area.

Fig. RR2: (a) TIRF image marked with 3 bright, 3 dim and 1 off-fibre area. (b) SRRF image. The scale bars shown in yellow measure 2.5 μm.

The SNR in Table RR1 was calculated using the formula,

$$SNR = \frac{\langle I_{fibre} \rangle - \langle I_{off-fibre} \rangle}{\sigma_{fibre}}$$

Table RR1: SNR of different fibres

Region	counts per pixel (Mean \pm SD)	SNR
D1	138.6 \pm 4.0	3.0
D2	136.7 \pm 3.8	2.7
D3	140.9 \pm 5.1	2.8
B1	160.8 \pm 7.1	4.8
B2	159.9 \pm 6.7	5.0
B3	175.6 \pm 10.5	4.7
Off-fibre	126.6 \pm 3.4	

We find that in Table RR1, as expected, the bright fibres have higher counts when compared to that of dim fibres. Empirically, we find that the SNR must be at least 3 for the fibre to be distinct on the SRRF image. A SNR of factor 3 indicates that the intensity of the fibre must be larger than the intensity off-fibre by at least 3 times its own standard deviation. We have added the following text to the manuscript.

“It is important to note that this thresholding based TIRF masking followed by D filtering strategy to correct for SRRF artefacts can be applied only on samples where a distinction can be made between the different regions of the cell. Empirically, we find that, the SNR must be at least 3 for efficient filtering using this approach.”

Note that the effects are hard to predict since the SRRF artifacts depend on the various settings and the specifics of the data (see for instance how easy it is to generate spurious features in Fig. S4 and Fig. S17).

>> We agree with the reviewer, artefacts are created by SRRF imaging and the amount of artefacts depend on the specific settings. Figs. S4 and S17 in the previous submission are representative SRRF images obtained using an EMCCD and sCMOS camera respectively. These images are raw SRRF images without any correction applied to them. These images were provided only to highlight the choice of time resolution used in this study. Since, we are currently performing a TIRF based correction on all SRRF images, we have now removed S17 from the submission. Fig. S4 is still provided to demonstrate the optimization steps in deciding spatial and temporal parameters for measurements. The SRRF images obtained after correction for Figs. S4 and S17 in previous submission are currently shown Figs. 2f and S15g respectively. Both these images clearly highlight the power of the correction strategy employed in this study since a considerable reduction in artefacts is observed. The Figs. 2f and S15g are shown here.

Fig. RR3: Representative EMCCD (Fig. 2f) and sCMOS (Fig. S15g) after correction are shown here. The scale bars shown in yellow measure 2.5 μ m.

While I support the spirit of the work of the authors, I can hardly escape from having serious reservations regarding the reliability of this strategy, and I am not convinced that it has the robustness desired for a general-purpose tool. If I was considering this tool for my own purposes then I would most likely stick to the TIRF and FCS data, and forget about the SRRF data entirely. And would I in fact lose anything much, since the FCS data is diffraction-limited anyway?

>> Imaging FCS data in our study is diffraction limited. We have now used only TIRF and diffusion coefficient to perform the filtering. The use of a TIRF mask removes off-fibre areas. Further refinement is provided by filtering based on diffusion coefficient retaining pixels that are consistent with diffusion on fibres. In the last step, we have used the diffusion map obtained after filtering to remove artefacts from the SRRF map.

We also observe that when we plot the number of particles across a fibre, the number of particles is lowest on the pixel identified as a fibre by SRRF. The number of particles per pixel is a proxy for the signal to noise ratio of an autocorrelation function. Typically, an increase in noise leads to an increase in the number of particles. The lowest number of particles on a SRRF pixel as measured by FCS on a SRRF's fibre containing pixel validates that the pixel identified as a fibre by SRRF is indeed a true positive-fibre and not a false positive-artefact. So in this case, we disagree with the reviewer that SRRF does not provide extra information. It clearly indicates the location of the fibre more precisely as corroborated by the SNR of FCS data.

The SRRF data is largely biased towards the bright features of the sample, so if desired I think I could do something similar but more robust by putting some more thought into the thresholding of the TIRF image (in the process also keeping those regions of the image that the SRRF calculation has improperly excluded).

>> We are sorry that the discussion of the article really surrounds mainly SRRF, which is not the topic of the article but represents one modality that can be used with our approach. We have shown this now with deconvolution and SOFI, both of which bring their own problems, but which also improve data extraction by correlating spatial and temporal data. We hope that clarifies our intentions. We also agree that other approaches that evaluate the structural information can be developed and used. To answer the question on bias, we agree that the SRRF data with gradient smoothing is largely biased towards the bright features of the sample. However disabling gradient smoothing (as in SRRF performed in this submission) leads to an improvement in fibre retention and a reduction in the bias towards bright samples.

The authors do comment on the thresholding with the TIRF image, though I find their reasoning unclear. They argue that "TIRF masking itself does not provide the localization of structures" (page 7 in the reply to the reviewers). But the whole point of any imaging is to provide the localization of structures! The authors continue that "TIRF can be used as a mask to remove SRRF artefacts". But how could it be used to decide which SRRF features are artifacts if it is incapable of providing the localization of structures?

>> We would like to apologize for our oversight. In our previous version, we had meant that TIRF masking does not provide localization beyond the diffraction limit. The sentence is not part of the manuscript anymore as we have removed SRRF filtering.

Sadly, upon taking all of this in consideration I regret that I cannot support publication of this revised work in Nature Communications.

Reviewer #2 (Remarks to the Author):

The authors have invested substantial efforts in improving the analysis workflow and in addressing my initial remarks and concerns. They present two convincing and self-sustained approaches to handle prominent artifacts introduced by SRRF analysis: i) subtracting an intensity-thresholded TIRF mask from the SRRF image and ii) removal of pixels from the SRRF mask which yielded outlying FCS observables, caused by bright and very slow diffusing aggregates. With the latter, the authors were able to show that the dynamic information obtained via FCS analysis can lead to more robust structural information obtained via SRRF, which also constitutes an improvement of SRRF on its own. This reviewer acknowledges the thorough discussions and elaborations added to the main text and the SI. In its revised form, an interested reader is able to grasp the methodology and the technical details as well as potential pitfalls regarding artifacts. Furthermore, the discussion of cellular data has largely benefited from additional experiments quantifying the effects of cholesterol removal and addition of EGF ligand to the system. I can now recommend the manuscript for publication in Nature Communications and look forward to seeing the method added to the toolbox of imaging-based biological research.

>> We would like to thank the reviewer for the positive comments.

Minor remarks:

Main

Line 321: ... the a more precise localization...

>> This sentence has been removed from this submission.

SI

Line 619: dynamics

>> This sentence has been removed from this submission.

Suppl. Note 3:

Unclear what is meant here: “Usually, the gradients are calculated using a 1×3 and 3×1 differentiation kernel pair in x and y directions respectively. Instead, the use of gradient smoothing leads to an increase in the size of the kernel pair used for calculating the gradient to 3×5 and 5×3 .” Pretty unclear what this means.

>> We are no longer using gradient smoothing in our SRRF analysis. This paragraph has been removed.

Reviewer #3 (Remarks to the Author):

The authors have well commented to most of my concerns as well have tackled those of the other referees – especially the other referees’ point of potential bias and reliability indeed

needed more attention. The manuscript has now significantly improved. I stick to my opinion that this manuscript deserves publication in NatComm.

>> We would like to thank Reviewer #3 for the positive comments.

One remaining issue from my side following my previous comment on the use of lipid domains. In my opinion the term “lipid domain” should be defined better – lipid domain on its own is too unspecific.

>>We have now defined “lipid domain” at the first use. The following sentence has been added to the manuscript.

“In this study, we use the term “lipid domains” to refer to 10-200 nm sized, dynamic, cholesterol and sphingolipid rich compartments in the cell membrane¹”

Points to be addressed

Dear Cara,

We have carefully evaluated the remarks raised by Reviewer 1 in an internal discussion. While we agree with most arguments, we still believe this work deserves publication in Nature Communications. However, we recommend refocusing the manuscript in line with some of the concerns. Before we suggest a structure, we respond to reviewer 1 point-by-point.

“The manuscript by Sankaran et al. has seen considerable work in these revisions. My main concern with the original version was the reliability of the SRRF imaging, an aspect that was not only identified by Reviewer 2 but that is now also explicitly mentioned by the authors, who agree that the technique is prone to introduce artifacts. The authors try to address this by mentioning that their approach could be combined with other computational frameworks, which is true, but such frameworks are not part of this contribution.”

This should be clarified by the authors.

>> We have now extended our approach by performing SOFI and deconvolution as well. Through these analyses, we would like to highlight the versatility of this technique that once the data is acquired using high spatiotemporal resolution, any computational microscopy algorithm can be applied post acquisition. SOFI and deconvolution images are shown in Fig. 2.

“The proposed solution is to adapt the analysis to mitigate the effects of such artifacts. This gives rise to a rather baroque strategy in which the FCS, SRRF, and TIRF data are repeatedly used to mask out various parts of the other images (figure S6). Such an analysis is unconventional, has many moving parts, and goes through great efforts for no reason other than to suppress the artifacts in the SRRF imaging.”

It is definitely worth communicating that from an TIRF-FCS dataset it is possible to take the SRRF option in order to obtain a super-resolution (SR) image at no cost (even if this technique is prone to artifacts). However, there is hardly any visible benefit on diffusion coefficient measurements using the SRRF image as an additional mask + D correction when compared to just TIRF masking + D correction (see Fig. S7a). This together with the fact that SRRF is prone to artifacts weakens the argument for the apparently essential requirement of

applying also a SRRF mask. In our opinion, the possibility to render an SR image from the same data set as complementary information does constitute novelty. Nevertheless even beyond, the authors identified a route to improve the truthfulness of the SRRF image: TIRF masking (straightforward) and filtering according to FCS observables. As stated in our last comments, this constitutes a novelty even for those interested in using SRRF on its own. A clear elaboration on those aspects would be a stronger storyline in our eyes.

>> We have now followed the advice of the reviewers. We use the TIRF and FCS filtering to remove artefacts from SRRF (Fig. 3) and highlight this in the article. We would also like to point out that FCS and SRRF data are mutually consistent as the actin fibre locations as defined by SRRF coincide with the FCS data of highest SNR. While this is not necessarily always true for all samples as it will depend on the background and the distribution of fluorescent molecules in the sample, in our case this is showing a direct correlation and underlines the point of mutual consistency (Fig. S12).

Fig. S12: Autocorrelation functions of pixels located on SRRF map: (a) The D_{TIRF} map ($D_{L,on(2)}^{TIRF} = 0.58 \pm 0.48 \mu\text{m}^2/\text{s}$) is shown at the top. The COV is 83%. (b) The enlarged view of the area labeled with the green box is shown below. The pixels bounded by the cyan lines are the pixels that overlap with SRRF image (white pixels in Fig. 2g). (c) The plot of $N_{L,on(2)}^{TIRF}$ versus the pixel type (F: fibre pixel, B: border pixel) for the three colored lines drawn in (b). Each colored line covers 3 pixels – a central pixel (on-fibre; Fig. S9-box 3) flanked by two border pixels (Fig. S9-box 1). All the pixels are plotted from the left to right for each line. (d) The ACFs of the blue line in (b). The ACF of the on-fibre pixel (pixel F in (c)) is shown in blue, while the ACFs of the border pixels are shown in grey. The scale bars shown in yellow measure 2.5 μm in images (a) and 250 nm in (b) respectively. Mean \pm SD of the D values are reported here.

“I have strong reservations about any technique where something as basic as acquiring an image requires such extensive ‘cleaning’ operations. This is the primary data that provides the

starting point for actual investigations, and reliability issues at that early stage will cast doubt onto the entire effort.”

Refocusing the storyline makes this point obsolete. However, in our opinion, many advanced imaging techniques nowadays require extensive post-processing and the raw data itself is often of limited value (e.g. SMLM). We do not share the strong reservations in that perspective.

>> We have now rewritten the article highlighting the use of Imaging FCS to perform SRRF correction. We agree that post-processing steps are necessary to maximize the extraction of information from the data. However with readily accessible open source plugins, as provided along with this article, facilitates multi-parametric data-evaluation, making this approach immediately applicable to a wide set of researchers.

“Note also that the SRRF procedure not only generates spurious signals, but also eliminates valid signals (see e.g. arrow 2 in Fig. RR1b). Such valid signals cannot be recovered by masking. In Fig. RR1 the authors focus on a specific parameter which appears to introduce artifacts, the 'gradient smoothing'. Disabling gradient smoothing may appear to reduce some of the artifacts, though at the end of the page the authors remark that "So gradient smoothing provides a better starting point for further refinement by FCS filtering". Why?”

This point also becomes obsolete when refocusing the manuscript. However, I agree that gradient smoothing appears somewhat opaque. As far as we understand it (but this point did not really come out too clear in the manuscript), gradient smoothing apparently improved spatial resolution of the SRRF image by a few nanometer. However, it seems that it eliminates valid signal, as Reviewer 1 correctly states. We would advise leaving out gradient smoothing since the small gain in spatial resolution is outweighed by the fact of losing valid signal.

>> We agree and have removed gradient smoothing from our analysis pipeline. All the images in our previous submission have been replaced with those calculated without gradient smoothing.

“Suppose that we give the authors the benefit of the doubt, and assume that the correction strategy described here works suitably well on this data. What if the end user acquires data that looks entirely different, and that does not have such a clear-cut distinction between fiber and non-fiber parts of the image, or where the signal levels are lower? Note that the effects are hard to predict since the SRRF artifacts depend on the various settings and the specifics of the data (see for instance how easy it is to generate spurious features in Fig. S4 and Fig. S17).”

Advanced microscopy techniques all have specific advantages with respect to certain type of specimens, their structures and features of interest. The authors should include a discussion on what limits the range of applicable samples.

>> We have evaluated the signal to noise ratio at two different conditions and have provided empirical estimates of SNR necessary for efficient filtering using the strategy suggested in this study. The following text has been added to the manuscript.

“It is important to note that this thresholding based TIRF masking followed by D filtering strategy to correct for SRRF artefacts can be applied only on samples where a distinction can be made between the different regions of the cell. Empirically, we find that, the SNR must be at least 3 for efficient filtering using this approach.”

“While I support the spirit of the work of the authors, I can hardly escape from having serious reservations regarding the reliability of this strategy, and I am not convinced that it has the robustness desired for a general-purpose tool. If I was considering this tool for my own purposes then I would most likely stick to the TIRF and FCS data, and forget about the SRRF data entirely. And would I in fact lose anything much, since the FCS data is diffraction-limited anyway? The SRRF data is largely biased towards the bright features of the sample, so if desired I think I could do something similar but more robust by putting some more thought into the thresholding of the TIRF image (in the process also keeping those regions of the image that the SRRF calculation has improperly excluded).”

See first point.

>> We have now used a TIRF mask as the first step in our filtering approach and we do not perform any SRRF based filtering.

“The authors do comment on the thresholding with the TIRF image, though I find their reasoning unclear. They argue that “TIRF masking itself does not provide the localization of structures” (page 7 in the reply to the reviewers). But the whole point of any imaging is to provide the localization of structures! The authors continue that “TIRF can be used as a mask to remove SRRF artefacts”. But how could it be used to decide which SRRF features are artifacts if it is incapable of providing the localization of structures?”

We agree with Reviewer 1, this sentence is clearly wrong and needs to be corrected.

>> We are sorry for this nonsensical statement. The sentence was meant to read “TIRF masking does not provide localization of structures beyond the diffraction limit and thus cannot filter the D map as efficiently as SRRF as seen from our results.” This sentence does not appear anymore in the revised manuscript.

“Sadly, upon taking all of this in consideration I regret that I cannot support publication of this revised work in Nature Communications.”

To summarize, we are still convinced that the work deserves publication in Nature Communications. Picking up several rightful concerns of Reviewer 1, we recommend the authors to reframe the story with focus on:

1. Imaging FCS with advanced thresholding (only TIRF mask and D correction) applied to solid biological data
2. SRRF SR images can be obtained from the imaging FCS data sets as complementary information
3. SRRF images are known to be prone to artifacts. FCS information can be used to provide a more truthful SR image using SRRF (we advise leaving out gradient smoothing)

>> We would like to thank the reviewers for this summary. We have rewritten the manuscript by utilizing only TIRF and FCS to perform filtering on the diffusion map. We then utilize this

diffusion map to remove artefacts from the SRRF map. We also validate the pixels identified as SRRF by utilizing information from FCS as true positive fibre pixels.

Reference

- 1 Pike, L. J. Rafts defined: a report on the Keystone Symposium on Lipid Rafts and Cell Function. *Journal of lipid research* **47**, 1597-1598, doi:10.1194/jlr.E600002-JLR200 (2006).

Reviewers' Comments:

Reviewer #2:

Remarks to the Author:

The authors have addressed all my concerns and the manuscript has significantly improved throughout the revisions. I fully support publication in Nature Communications.

Minor remarks:

Line 224: "Thickness (FWHM of the Gaussian fit) of the actin fibre before SRRF (magenta line in (a)) = 392 nm". The magenta line is labeled as TIRF image and it should be also described as such here. Since multiple computational postprocessing steps are now included (which I appreciate), it is misleading what before SRRF means, e.g. SOFI order 4?

Line 254: Caption Figure 3: "Schematic outlining the strategy of using SRRF image and D map to correct each other" – I think the wording is misleading (and originates from former storyline) since the SRRF image is NOT used to correct the D map, i.e. each other is wrong. Please describe precisely.

Response to Reviews

Reviewer #2 (Remarks to the Author):

1. The authors have addressed all my concerns and the manuscript has significantly improved throughout the revisions. I fully support publication in Nature Communications.

>> We would like to thank the reviewer for the positive comments.

Minor remarks:

2. Line 224: “Thickness (FWHM of the Gaussian fit) of the actin fibre before SRRF (magenta line in (a)) = 392 nm”. The magenta line is labeled as TIRF image and it should be also described as such here. Since multiple computational postprocessing steps are now included (which I appreciate), it is misleading what before SRRF means, e.g. SOFI order 4?

>> Before SRRF meant the original TIRF image. The legend has now been corrected.

“Thickness (FWHM of the Gaussian fit) of the actin fibre in TIRF image (magenta line in (a)) = 392 nm.”

3. Line 254: Caption Figure 3: “Schematic outlining the strategy of using SRRF image and D map to correct each other” – I think the wording is misleading (and originates from former storyline) since the SRRF image is NOT used to correct the D map, i.e. each other is wrong. Please describe precisely.

>>We would like to thank the reviewer for pointing this out. The legend now reads

“Schematic outlining the strategy to improve the accuracy and precision of the D map, and its further use to correct the SRRF image”.